# AVEX: What Matters for Animal Vocalization Encoding

**Marius Miron**[†][⋆]
Earth Species Project

**David Robinson**[†][⋆]
Earth Species Project

**Milad Alizadeh**[†]
Earth Species Project

**Ellen Gilsenan-McMahon**[†]
Earth Species Project

**Gagan Narula**[†]
Earth Species Project

**Emmanuel Chemla**
Earth Species Project

**Maddie Cusimano**
Earth Species Project

**Felix Effenberger**
Earth Species Project

**Masato Hagiwara**
Earth Species Project

**Benjamin Hoffman**
Earth Species Project

**Sara Keen**
Earth Species Project

**Diane Kim**
Earth Species Project

**Jane Lawton**
Earth Species Project

**Jen-Yu Liu**
Earth Species Project

**Aza Raskin**
Earth Species Project

**Olivier Pietquin**[†][‡]
Earth Species Project

**Matthieu Geist**[†][‡]
Earth Species Project

## Abstract

Bioacoustics, the study of sounds produced by living organisms, plays a vital role in conservation, biodiversity monitoring, and behavioral studies. Many tasks in this field, such as species, individual, and behavior classification and detection, are well-suited to machine learning. However, they often suffer from limited annotated data, highlighting the need for a general-purpose bioacoustic encoder capable of extracting useful representations for diverse downstream tasks. Such encoders have been proposed before, but are often limited in scope due to a focus on a narrow range of species (typically birds), and a reliance on a single model architecture or training paradigm. Moreover, they are usually evaluated on a small set of tasks and datasets. In this work, we present a large-scale empirical study that covers aspects of bioacoustics that are relevant to research but have previously been scarcely considered: training data diversity and scale, model architectures and training recipes, and the breadth of evaluation tasks and datasets. We obtain encoders that are state-of-the-art on the existing and newly proposed benchmarks. We also identify *what matters* for training these encoders, such that this work can be extended when more data are available or better architectures are proposed. Specifically, across 26 datasets with tasks including species classification, detection, individual ID, and vocal repertoire discovery, we find that self-supervised pre-training followed by supervised post-training on a mixed bioacoustics + general-audio corpus yields the strongest in- and out-of-distribution performance. We show the importance of data diversity in both stages. To support ongoing research and applications, we release the model checkpoints as well as the Animal Vocalization Encoder library AVEX (an API for model loading and inference, and a Python-based system for training and evaluating bioacoustics representation learning models)[1].

---

[†]Core authors.
[⋆]Equal contribution, corresponding authors: {marius, david}@earthspecies.org.
[‡]Equal senior contribution.
[1]https://projects.earthspecies.org/avex/

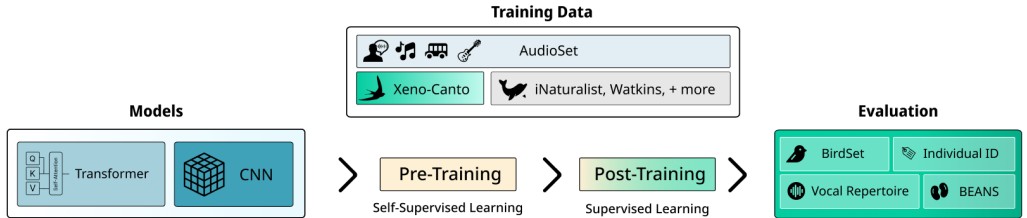

Figure 1: Our empirical study diagram, assessing (1) models, (2) training data, (3) training paradigms, and proposing an (4) extended evaluation data and methodology.

# 1 INTRODUCTION

Bioacoustics is the study of animal sound production and perception (Bradbury & Vehrencamp, 1998). It is a crucial component for understanding animal behavior (Fischer et al., 2013), for biodiversity monitoring and conservation efforts (Rutz et al., 2023; Stevens et al., 2024), and for modeling the mechanisms underlying animal communication (Bradbury & Vehrencamp, 1998). A variety of common tasks in bioacoustics are used to support these efforts: sound event detection or classification of species, individuals, call-types, and behaviors. All of these tasks are well-suited for a machine learning approach. Machine learning and deep learning are now commonly used for bioacoustics (Stowell, 2022), and have enabled discoveries such as the use of specialized vocalizations for labeling conspecifics in marmosets (Oren et al., 2024) or elephants (Pardo et al., 2024). However, due to unavoidable challenges in data collection and annotation, these studies generally rely on small datasets strongly labeled on a few species and individuals (Stowell, 2022). The resulting bioacoustic machine learning models are then usually designed for specific tasks and species (Dufourq et al., 2021; Cauzinille et al., 2024), limiting their generalizability.

However, large amounts of unannotated or weakly labeled bioacoustic data are recorded regularly, especially through Passive Acoustic Monitoring (PAM) (Gibb et al., 2019), and citizen science platforms such as Xeno-Canto (Vellinga & Planqué, 2015) or iNaturalist (Chasmai et al., 2024). These data can be leveraged to train a bioacoustic encoder, which can then be deployed in downstream tasks, as bioacoustic features (*e.g.* for linear probing or clustering) or finetuning the whole model, among other options. Two such classic and state-of-the-art bioacoustic encoders are BirdNet (Kahl et al., 2021) and Perch (Ghani et al., 2023) which have been applied to downstream application tasks such as multi-taxa species retrieval and detection (Pérez-Granados, 2023; Dumoulin et al., 2025; Ghani et al., 2023).

Other bioacoustic encoders have been proposed, to be reviewed in the next section. They have in common a supervised learning approach, usually limited to a single taxonomic group, with notable exceptions including SurfPerch (Williams et al., 2024) and recently the models of iNatSounds (Chasmai et al., 2025). Moreover, they evaluate the quality of the learned representations on a limited set of downstream tasks and datasets. Typically they solely evaluate on species classification, with their training and test data containing the same species, often with an out-of-distribution effect, as training datasets typically consists of focal recordings, while evaluation datasets are soundscape recordings (Rauch et al., 2025b). In contrast, real-world bioacoustic applications require encoders that generalize effectively across diverse species and tasks, often beyond those explicitly seen during training. For example, researchers may need to recognize previously unobserved species, identify individual animals from limited vocal data, or characterize animal vocal repertoires without extensive annotations. Evaluating models on such diverse and realistic scenarios is critical, yet building and measuring the performance of encoders that generalize across these conditions remains underexplored in current works.

Our main contribution is an empirical study assessing what components matter most for training a generalizable bioacoustic encoder. We systematically investigate (1) model architectures, (2) data-mixes, and (3) training paradigms under a (4) broadened evaluation methodology. **(1)** Specifically, in terms of models we compare CNN-based (LeCun et al., 1989) and transformer-based (Vaswani et al., 2017) architectures alongside their associated learning approaches: supervised and self-supervised.

(**2**) On the data mix aspect, we train and evaluate across a broader and more taxonomically diverse bioacoustic dataset than previous work, examining the impact of incorporating general audio data such as AudioSet (Gemmeke et al., 2017). (**3**) Additionally, we explore sequential training paradigms ("training recipes"), pre-training and post-training, including self-supervised and supervised learning, and assess the influence of non-bioacoustic audio data at different training stages. (**4**) By evaluating these models across established benchmarks BEANS (Hagiwara et al., 2023), and BirdSet (Rauch et al., 2025b) alongside newly curated datasets assessing generalization to challenging real-world tasks, we provide a clearer picture of the conditions that enhance bioacoustic representation learning. We find that under comparable training conditions, self-supervised models achieve strong out-of-distribution generalization yet under-perform supervised models on in-distribution tasks, and that incorporating general audio into bioacoustic training significantly improves model transferability. Sequential self-supervised and supervised learning yields strong performance both in and out-of-distribution. Leveraging these insights, we propose a set of training recipes and models that achieve state-of-the-art results overall on our extensive evaluation benchmark, offering a versatile encoder for bioacoustic research.

## 2 RELATED WORK

**Self-Supervised Audio Encoders.** An extensive number of works propose audio and speech encoders, most of them transformer-based, such as Wav2vec (Baevski et al., 2020), HuBERT (Hsu et al., 2021), AudioMAE (Huang et al., 2022), BEATs (Chen et al., 2023) or EAT (Chen et al., 2024). Several bioacoustic-specific encoders have also been developed: BirdNet (Kahl et al., 2021) and Perch (Ghani et al., 2023) build upon an EfficientNet (EffNet) architecture (Tan & Le, 2021), a CNN-based vision neural network pre-trained on ImageNet (Russakovsky et al., 2015) taking audio spectrograms as input. Transformer-based bioacoustic models include AVES (Hagiwara, 2023) based on HuBERT, Animal2Vec (Schäfer-Zimmermann et al., 2024) based on data2vec (Baevski et al., 2023), BirdMAE (Rauch et al., 2025a) based on AudioMAE with modified decoder architecture, and TweetyBert (Vengrovski et al., 2025) inspired by BERT (Devlin et al., 2019). However, within bioacoustics these approaches lack systematic comparison across different architectures with standardized pipelines. In contrast, we compare with a wide range of encoder baselines and aim to have a fair comparison across different architectures, with minimal changes and near-identical pipelines.

**Text-Informed Audio Models and Their Relation to Bioacoustic Encoders.** Audio encoders are key in training text-informed models for bioacoustics. BioLingual (Robinson et al., 2024) learns a common representation for text and bioacoustics, inspired by CLAP-LAION (Wu et al., 2023). It allows to perform tasks like zero-shot species classification or text-to-audio retrieval, while a bioacoustic encoder requires further learning (like linear probing). NatureLM-audio (Robinson et al., 2025) is a large audio-language model for bioacoustics, adding audio as input to a Llama 3 model (Grattafiori et al., 2024) through a BEATs audio encoder and a Q-former (Li et al., 2023). In this work, we focus on bioacoustic encoders, and this line of work is complementary to text-informed or large language models, in the sense these models could benefit from a better encoder. It can also be considered as a "post-training" stage for any bioacoustic encoder, that can then be extracted to be used in downstream bioacoustic tasks, for example doing linear probing. As a baseline, we consider extracting the BEATs encoder of NatureLM-audio, which was unfrozen during training on large-scale bioacoustic data.

**Data mixing in bioacoustics training.** In terms of data composition, existing bioacoustic encoders typically use limited data sources: BirdNet and Perch are post-trained on bird data mostly from Xeno-Canto, while Williams et al. (2024) extended Perch to Surfperch by adding coral reef bioacoustic data. Animal2Vec focuses specifically on meerkats data, and TweetyBert on canary song. General audio encoders are evaluated in extensive audio (Turian et al., 2022) and speech benchmarks (Yang et al., 2021) but contain little bioacoustic data in their training mix. While these general-purpose encoders can be used for bioacoustic tasks, Sarkar & Doss (2025) found that pre-training on bioacoustic data provides only marginal improvements, though other studies reach different conclusions (Ghani et al., 2023; Rauch et al., 2025a). Our work differs by considering larger and more taxonomically diverse bioacoustic training data, examining the impact of adding general audio data alongside bioacoustic data, and evaluating the impact of data mix under fair settings rather than only evaluating pre-existing models.

**Training Paradigms: Self-Supervised, Supervised, and Two-Stage Approaches.** Regarding the training paradigm, current bioacoustic modeling approaches use either self-supervised learning (AVES, Animal2Vec, BirdMAE, TweetyBert) or supervised learning (BirdNet, Perch, Surfperch) exclusively. However, no existing work systematically explores the combination of both paradigms or examines the impact of including bioacoustic data at different training stages. We address this gap by considering the combination of both self-supervised and supervised learning. To that extent, our pre- and post-training formalization may be seen as a form of curriculum learning (Bengio et al., 2009) with two stages, similar to the iterative training of BEATs and commonly used in training LLMs (Robinson et al., 2025).

**Evaluation Limitations in Bioacoustic Benchmarks.** Bioacoustic encoders have been evaluated primarily in the context of species classification and detection (Rauch et al., 2025b; Hamer et al., 2023; Ghani et al., 2023; Chasmai et al., 2025; Kather et al., 2025) while other important tasks such as vocal repertoire discovery(Anikin et al., 2018) or individual identification(Stowell et al., 2019) have been scarcely addressed. These tasks, which are critical to the study of animal communication yet lack large-scale annotated data, are a natural test-bed for generalization of learned representations. Research on these two topics has so far used a limited number of private datasets or has not compared with state-of-the-art bioacoustic encoders (Best et al., 2023; Nolasco et al., 2025; Stowell et al., 2019; Wierucka et al., 2025). To give a broader overview of the capabilities of a bioacoustic encoder, we address these limitations by adding 8 public datasets, not considered in any previous benchmark. Further, related work from Kather et al. (2025) and from Best et al. (2023) has gained insight into bioacoustic encoders by analyzing their embeddings with clustering metrics, including a qualitative finding that self-supervised encoders better generalized from birds to frogs. We introduce a similar evaluation methodology, enhancing BEANS (Hagiwara et al., 2023) and BirdSet (Rauch et al., 2025b) with clustering and retrieval metrics, scaling a related analysis from two datasets to twenty-six. For a comparison with current bioacoustics benchmarks, we present Table 4, Appx A, which includes training and evaluation configurations.

## 3 METHODS

This section provides the details of our empirical study which we summarize in Figure 1.

### 3.1 TRAINING DATA

State-of-the art bioacoustic encoders are either trained in a self-supervised manner on large datasets comprising general audio (Hagiwara, 2023) and birds (Rauch et al., 2025a), or in a supervised manner to predict the species in focal recordings of birds (Rauch et al., 2025b; Hamer et al., 2023; Van Merriënboer et al., 2024; Rauch et al., 2025a; Ghani et al., 2023). We extend these paradigms by comparing the self-supervised and supervised learning on both types of data, general audio and bioacoustic data, both comprising labels. The data are used for two approaches, self-supervised learning (in which case the labels are ignored) and supervised learning. The general audio dataset is AudioSet (Gemmeke et al., 2017) comprising labels for sound event detection within the AudioSet ontology. With respect to bioacoustics data, we compile a large dataset from multiple sources, including Xeno-canto (Vellinga & Planqué, 2015), the largest source of bird signals, iNaturalist (Chasmai et al., 2024), Animal Sound Archive (Museum für Naturkunde Berlin, 2023), which includes diverse taxa, and the Watkins Marine Mammal database "all cuts" (Sayigh et al., 2016) offering the most diverse collection of marine mammal signals. Outside of our core training mix, we consider additional bioacoustic soundscape datasets to study their effect on the learned representations, in particular WABAD (Pérez-Granados et al., 2026) and Sapsucker Woods (Kahl et al., 2022a). To join diverse bioacoustic datasets, we curate species' scientific names and link all species to a common taxonomic backbone (GBIF) (Telenius, 2011). We summarize the training data in Table 1.

We train models with noise augmentation (see Section 3.3) using non-animal environmental sounds from the following datasets: ShipsEar (Santos-Domínguez et al., 2016), Deepship (Irfan et al., 2021) and Orcalab (Poupard et al., 2020), FSD50K (Fonseca et al., 2021), Urbansound (Salamon & Jacoby, 2014), TUT2016 (Mesaros et al., 2016b), IDMT (Abeßer et al., 2021), Demand (Thiemann et al., 2013), and Wham (Wichern et al., 2019).

Table 1: Datasets used in pre-training and post-training. $^a$ denotes datasets used solely in ablations.

| Dataset | # Hours | Description |
|---|---|---|
| `AudioSet` (Gemmeke et al., 2017) | 5700 | general audio |
| `Xeno-canto` (Vellinga & Planqué, 2015) | 10416 | birds |
| `iNaturalist` (Chasmai et al., 2024) | 1539 | diverse taxa |
| `Watkins` (Sayigh et al., 2016) | 27 | marine mammals |
| `Animal Sound Archive` (Museum für Naturkunde Berlin, 2023) | 78 | diverse taxa |
| `Sapsucker Woods`$^a$ (Kahl et al., 2022a) | 285 | birds |
| `WABAD`$^a$ (Pérez-Granados et al., 2026) | 84 | birds |

## 3.2 Pre-existing Encoders

We consider various pre-existing encoders, as baselines and for further benchmarking and analysis. First, we consider general audio encoders, more specifically BEATs (Chen et al., 2023) and EAT (Chen et al., 2024), BEATs because it is a state-of-the-art encoder, and EAT because we will modify its self-supervised training recipe; Chen et al. (2023) do not provide the training code, only trained checkpoints, and EAT is a good and (fully) open-sourced model.

We also include bioacoustic encoders as baselines, namely BirdNet (Kahl et al., 2021) and Perch (Ghani et al., 2023) as state-of-the-art baselines. In addition, we evaluate SurfPerch (Williams et al., 2024) because it uses more diverse taxa in training. We also consider AVES (Hagiwara, 2023) and BirdMAERauch et al. (2025a) as a representative self-supervised models for bioacoustics.

For the last baseline, we extract the BEATs encoder from NatureLM-audio (Robinson et al., 2025), that we call NatureBEATs, as a representative of an unorthodox post-trained encoder. Comparing it to BEATs can provide clues about the influence of text-audio training and of post-training with bioacoustic data, in addition to the experiments to be described now.

## 3.3 Proposed Models and Training Recipes

We provide a summary of the models we train according to data used in pre- and post-training in Table 2. As explained before, both BirdNet and Perch build upon an EffNet post-trained on (mostly) Xeno-Canto, using a multi-label supervised learning loss. To mimic this approach, we also consider an EffNet architecture, with a checkpoint pre-trained on ImageNet, which we post-train with supervised learning and a binary cross-entropy loss. To assess the utility of using bioacoustic data, possibly complemented by general audio data, we consider post-training this model on bioacoustic data only, on AudioSet only, or on both. The version post-trained solely on bioacoustic data is reminiscent of BirdNet or Perch, even if it is not an apple-to-apple comparison (each model uses a different bioacoustic dataset, normalization method, sampling rate, spectrogram parameters, and augmentation details). We take advantage of the efficiency of the architecture to test various data-mixes, including the addition of soundscape data, and the ablation of large taxonomic subgroups such as birds, whales, or all non-birds. The results for this ablation are presented in Figures 4 and 5 in the Appendix.

Our approach does not rely specifically on the EffNet architecture. Therefore, we propose to do the post-training of another, transformer-based, audio encoder. We chose BEATs, as it is a state-of-the-art encoder, pre-trained on speech and general audio data with a self-supervised approach. We post-train it in a supervised manner on our bioacoustic data only, or on both bioacoustic data and AudioSet. We exclude post-training BEATs on AudioSet only as this corresponds to the public checkpoint BEATs (SFT).

We can also envision modifying the self-supervised pre-training phase. Unfortunately, the training code for BEATs is not available and is not straightforward to reproduce. Therefore, we turn to the EAT model (Chen et al., 2024), which also provides good results, is fully open-source, and has the advantage of being fast to train. We consider pre-training EAT without modifying its self-supervised learning approach (which is a mix of teacher distillation and reconstruction of masked patches of the spectrogram), on our bioacoustic data only, on AudioSet only, and on both. Then, we consider not post-training this model (to assess if post-training is useful, especially given that it is the same

dataset, up to ignoring or not the labels), and also post-training it as before (bioacoustic data only, or bioacoustic data plus AudioSet). Similar to BEATs, we do not post-train EAT on AudioSet only as this matches an existing checkpoint, "EAT (SFT)". We use the "EAT-all" SSL checkpoint as the basis for post-training.

For a fair comparison with pre-existing models we train and evaluate all our models at 16kHz, whilst we acknowledge that some species contain important auditory information above 8kHz and we plan to extend this study in future work. We evaluate Perch and BirdNet at their proposed sample rates 32kHz and 48 kHz, and to their advantage, they observe a broader frequency spectrogram than our base EffNet.

To increase generalization and robustness to noise of the learned representation we found it important to use two augmentations. Namely, during pre-training and post-training we add noise randomly with a probability of 0.5 at a random signal-to-noise ratio (SNR) sampled from a uniform distribution between $-10$dB and 20dB using the datasets introduced in Section 3.1. During post-training, with probability 0.5 we linearly mix random pairs of audio clips within a batch and set the target to the union of their labels (element-wise OR).

Table 2: Pre- and post-training datasets with resulting model checkpoints. [†] Indicates checkpoints released by prior work. "AS"=AudioSet. All post-trained EAT models use the EAT-all checkpoint as the base.

| Arch. | Pre-train data | Pre-trained checkpoint | Post-train data | Resulting checkpoint(s) |
|---|---|---|---|---|
| EffNetB0 | ImageNet[†] | – | Bio / All / AS | EffNetB0-bio, EffNetB0-all, EffNetB0-AS |
| BEATs | AS[†] | BEATs (pre)[†] | Bio / All / AS | sl-BEATs-bio, sl-BEATs-all, BEATs (SFT)[†] |
| EAT | AS[†] | EAT-base (pre)[†] | AS | EAT-base (SFT)[†] |
| EAT | Bio / All / AS | EAT-bio, EAT-all, EAT-AS | Bio / All | sl-EAT-bio, sl-EAT-all |

### 3.4 EVALUATION SETUP: DATA, TASKS, AND METRICS

**Evaluation Tasks.** We consider two tasks commonly addressed in the literature: *classification* on audio excerpts into discrete category labels and *detection* of events in longer audio files. We employ three distinct evaluation setups to assess model representations comprehensively. *Linear probing* trains a linear classifier on train split using time-averaged embeddings from the final layer of the model (excluding classification heads), and then evaluates this probe on the test split. *Retrieval evaluation* directly investigates model embedding spaces by treating each test set item as a query and ranking remaining items by cosine similarity. *Clustering evaluation* performs K-means with known cluster counts on single-label datasets, evaluating similarity to ground truth classes using normalized mutual information (marked as NMI).

**Evaluation Paradigms: Probing, Retrieval, and Clustering.** Current bioacoustics benchmarks consider either an in-domain multiple taxa setup, such as BEANS (Hagiwara et al., 2023) and (Robinson et al., 2024), or out-of-domain bird species detection, such as BIRB (Hamer et al., 2023) and BirdSet (Rauch et al., 2025b), focusing on generalization. Here we extend these benchmarks in multiple ways. **(1)** We expand the tasks to include individual identification and vocal repertoire discovery, along with the curation of eight additional benchmark datasets. This extends the evaluation of bioacoustic encoders from a focus on species classification of birds to a focus on generality across multiple tasks and taxa. **(2)** Extending the work of Kather et al. (2025), we also augment existing benchmarks with clustering and retrieval metrics, allowing us to examine a model's embedding spaces directly. Each of these benchmarks, tasks, and metrics is directly tied to downstream research or conservation applications and allows us to analyze both pre-existing and new models from a different perspective.

**Evaluation datasets.** We present the datasets and tasks we consider as part of our evaluation in Table 4, Appendix A. We evaluate on the classification and detection tasks in the already existing benchmarks BEANS (Hagiwara et al., 2023) and BirdSet (Rauch et al., 2025b). For BEANS we

follow the official train/validation/test splits in the original benchmark. We exclude the auxiliary dataset SpeechCommands from BEANS but we include ESC-50 because it still may be useful for conservation tasks relevant to bioacoustics e.g. habitat classification, poaching monitoring. For BirdSet, we match their "Dedicated Train" setup, considering separate train and test splits for each of the datasets. Accordingly, we derive the validation dataset as a stratified split on the species from the train with $0.8, 0.2$ ratios and seed $42$. We exclude SSW (Kahl et al., 2022b) from BirdSet evaluation as we consider it within some of our data-mix. We formulate detection similarly to BEANS and BirdSet as segment-based multi-class classification and we use segment-based sound event detection metrics to evaluate it (Mesaros et al., 2016a), allowing for the negative class (no class detected). We leave frame-based (Mesaros et al., 2016a) or event-based (Mahon et al., 2025) temporally-strong detection for future work. Importantly, solely for BirdSet which contains an important covariate shift from train (focal recordings) to test (soundscapes) we use the same noise and mixup augmentations in pre-training and post-training.

**New Evaluation Datasets: Individual Identification and Vocal Repertoire Evaluation.** For the two new evaluation tasks we compile public datasets and create label-stratified train/validation/test splits (seed $42$, ratios $0.6/0.2/0.2$). The Individual Identification task (Stowell et al., 2019; Fukushima et al., 2015) is a supervised single-label classification problem over individuals of the same species. The Vocal Repertoire Discovery setting (Elie & Theunissen, 2016; Mumm & Knörnschild, 2014; Cohen et al., 2020; Palmer et al., 2025) evaluates how well embeddings discriminate between the different call types within a species' vocal repertoire. We treat this as a structure-recovery problem with *known* $K$ (the number of annotated call types): no probes are trained; labels are used only as a reference to assess representation quality via (i) clustering (K-means with $K$ equal to the number of call types; scored by normalized mutual information, NMI) and (ii) audio-to-audio retrieval within call type (ROC AUC, R-AUC). This matches repertoire discovery when $K$ is known. Supervised call-type classification is also a useful formulation of the task, but performance on several datasets was near-ceiling while others were too small for quality splits, so we exclude linear probing for this task.

**Probing Evaluation Setup** To directly evaluate learned representations, we use linear probing on time-averaged embeddings rather than full fine-tuning as Hagiwara (2023); Rauch et al. (2025b). This avoids confounding effects from model size differences, ensuring fair comparison between CNN and transformer representations. While performance may vary across embedding layers, all models show similar trends (Cauzinille et al., 2024; Sarkar & Doss, 2025). Evaluating all layers would exponentially increase computational cost, so we extract embeddings from the final layer (excluding classification heads) and leave comprehensive layer analysis for future work. Considering a frozen representation (rather than fully finetuning) is also especially important for many downstream applications, as it allows notably precomputing the embeddings (Dumoulin et al., 2025). In terms of hyperparameters we use a learning rate of 1e-4, weight decay of 0.1, batch size of 32, and 900 epochs.

**Metrics.** For all tasks other than Vocal Repertoire Discovery we report classic performance metrics on linear probing (top-1 accuracy for single-label tasks, macro-averaged mean average precision for multilabel tasks). To evaluate retrieval, we consider each item in the test set as a query. We rank the remaining elements of the test set according to their cosine similarity with the query under the model's embedding function, and evaluate the ordering. For single-label tasks (BEANS Classification, Individual ID, and Vocal Repertoire), we consider items as relevant to the query if they share the same label as the query. For multi-label tasks (BEANS Detection, BirdSet) we consider items to be relevant if they share at least one label with the ground-truth item. We exclude queries with no labels, while ground-truth items with no labels are included in the evaluation as negatives. We measure this ranking using ROC AUC as the primary metric (marked as R-AUC with R for retrieval). For Individual Identification evaluation datasets Pipit and Chiffchaff (Stowell et al., 2019), the training set follows an intentionally different background noise distribution than the test set following original design in Stowell et al. (2019). To evaluate generalization in this scenario, we report ROC AUC by using each example in the train set as a query, rather than each example in the test set as a query. For single-label datasets in all benchmarks, we also consider a clustering task. For each evaluation set, we perform K-means clustering given a known number of clusters (labels). We evaluate similarity of these clusters compared to the known class groups by measuring normalized mutual information (marked as NMI) between the two. These two additional evaluations offer insight into the pre-trained embeddings of a model, but are also relevant to downstream tasks in bioacoustics,

namely audio-to-audio retrieval (Hamer et al., 2023) (retrieval) and repertoire discovery (Best et al., 2023) (clustering) or call-type classification. For vocal repertoire discovery, we use clustering and retrieval as the primary evaluations. All metrics are formalized in the Appendix in B.2.

# 4 RESULTS

## 4.1 WHAT DATA MATTERS IN POST-TRAINING EFFICIENTNET

We present results for our EffNet models aggregated across benchmarks in Table 3, the shaded rows. We compare to Perch, SurfPerch, and BirdNet - all EffNet-based and considered state-of-the-art bioacoustic encoders. The results highlight the value of our diverse curated bioacoustic datasets, with our best EffNet model outperforming on eight of ten metrics. We observe a consistent performance gain of including general audio in the data-mix, transferring across focal classification, multi-label classification on soundscapes, vocal repertoire, and individual ID tasks. Consistent with other works (Ghani et al., 2023), supervised training on general audio alone transfers poorly compared to bioacoustic data. Interestingly, this holds for the newly benchmarked Vocal Repertoire and Individual ID tasks, suggesting large-scale species-prediction is an effective approach for transfer to these tasks, which are often studied independently. We further characterize what supervised data transfers to which tasks and species through further ablations in the Appendix in Figure 3.

## 4.2 SELF-SUPERVISED PRE-TRAINING HELPS OUT-OF-DISTRIBUTION

Comparing the results of EAT models trained with self-supervised learning, we find a strong effect of including general audio in the data-mix, with the model trained with the addition of AudioSet significantly outperforming the bioacoustics-only model across tasks (Figure 2a).

We further compare our supervised and self-supervised models (Figure 2b) alongside existing models, and alongside post-trained SSL models, discussed in the following section. For this comparison we want to see how discriminative raw representations from the models are with respect to the target datasets, and we use ROC AUC as a training-free metric. Compared across benchmarks, this analysis should give an idea about how generalizable the embedding space is. In this analysis we trade-off experimental control for scale, using large datasets. Running the same analysis on datasets where we control for species distribution, noise conditions, and other confounders should give a better explanation on how robust these representations are. Because we are not aware of any dataset that offers this controlled conditions we leave this for future work.

While supervised models excel in tasks closely matching their training distribution, self-supervised models demonstrate superior generalization capabilities on out-of-distribution tasks. Specifically, we find when generalizing from BEANS classification (typically focal recordings) to BEANS Detection (entirely soundscape recordings) the self-supervised models drop on average only 0.01 retrieval ROC AUC compared to a drop of 0.09 retrieval ROC AUC for the supervised models. The strength of the effect is sufficient that the best pure self-supervised model, the pre-trained BEATs, outperforms the strongest pure supervised models by retrieval on BEANS Detection. This finding underscores the strong potential of self-supervised learning in bioacoustics, where supervised learning is still considered state-of-the-art, yet models are challenged by huge distribution shifts between training and deployment (Hamer et al., 2023).

## 4.3 POST-TRAINING RECIPES FOR SELF-SUPERVISED BACKBONES

We show the results of self-supervised backbones post-trained on our bioacoustic dataset, compared to state-of-the-art bioacoustic encoders in Table 3. Our best post-trained models outperform overall, achieving state-of-the-art performance across the established BEANS Classification, BEANS Detection, and BirdSet benchmarks, outperforming both their self-supervised base models and supervised baselines. On the newly-proposed Vocal Repertoire Discovery and Individual Identification benchmarks, the post-trained models maintain competitive performance, but the newly-trained EffNet on mixed bioacoustic and general audio data performs best, and BirdNet performs strongest by retrieval on Individual Identification. With respect to the the data-mix, the BEATs model trained on the mix of general audio and bioacoustic audio outperforms overall, while the mixed training has a more variable effect on EAT. We additionally find that post-training retains some of the out-of-distribution

| Model | BEANS Classification | | | BEANS Detection | | BirdSet | | Individual ID | | Vocal Repertoire | |
|---|---|---|---|---|---|---|---|---|---|---|---|
| | *Probe* | *R-auc* | *C-nmi* | *Probe* | *R-auc* | *Probe* | *R-auc* | *Probe* | *R-auc* | *R-auc* | *C-nmi* |
| **BEATS (SFT)**[SSL] | 0.724 | 0.739 | 0.504 | 0.339 | 0.692 | 0.101 | 0.675 | 0.375 | 0.602 | 0.755 | 0.485 |
| **BEATS (pretrained)**[SSL] | 0.774 | 0.734 | 0.542 | 0.381 | 0.722 | 0.129 | 0.686 | 0.380 | 0.637 | 0.775 | 0.498 |
| **EAT-base (pretrained)**[SSL] | 0.679 | 0.675 | 0.424 | 0.252 | 0.692 | 0.104 | 0.650 | 0.363 | 0.623 | 0.768 | 0.467 |
| **EAT-base (SFT)**[SL-SSL] | 0.758 | 0.748 | 0.478 | 0.358 | 0.714 | 0.143 | 0.676 | 0.418 | 0.632 | 0.817 | 0.527 |
| **Bird-AVES-biox-base**[SSL] | 0.705 | 0.646 | 0.410 | 0.340 | 0.689 | 0.092 | 0.670 | 0.402 | 0.622 | 0.726 | 0.453 |
| **NatureBEATs**[SL-SSL] | 0.804 | 0.774 | 0.560 | 0.385 | 0.724 | 0.223 | 0.723 | 0.410 | 0.645 | 0.811 | 0.552 |
| **Bird-MAE-Huge**[SSL] | 0.766 | 0.674 | 0.432 | 0.354 | 0.680 | 0.168 | 0.636 | 0.404 | 0.637 | 0.812 | 0.485 |
| **SurfPerch**[SL] | 0.760 | 0.745 | 0.484 | 0.301 | 0.664 | 0.160 | 0.694 | 0.457 | 0.656 | 0.751 | 0.492 |
| **BirdNet**[SL] | 0.796 | 0.772 | 0.523 | 0.392 | 0.687 | N/A | N/A | 0.472 | **0.708** | 0.795 | 0.545 |
| **Perch**[SL] | 0.768 | 0.759 | 0.478 | 0.368 | 0.674 | 0.233 | 0.656 | 0.530 | 0.705 | 0.758 | 0.493 |
| **EffNetB0-AudioSet**[SL] | 0.651 | 0.721 | 0.486 | 0.246 | 0.670 | 0.098 | 0.655 | 0.397 | 0.612 | 0.760 | 0.481 |
| **EffNetB0-bio**[SL] | 0.786 | 0.799 | 0.563 | 0.365 | 0.695 | 0.279 | 0.704 | 0.457 | 0.683 | 0.806 | 0.568 |
| **EffNetB0-all**[SL] | 0.800 | 0.809 | 0.584 | 0.362 | 0.712 | 0.279 | 0.707 | **0.531** | 0.701 | **0.830** | **0.582** |
| **EAT-AS**[SSL] | 0.704 | 0.714 | 0.473 | 0.311 | 0.704 | 0.125 | 0.685 | 0.362 | 0.627 | 0.801 | 0.533 |
| **EAT-bio**[SSL] | 0.692 | 0.671 | 0.410 | 0.311 | 0.679 | 0.143 | 0.631 | 0.378 | 0.627 | 0.757 | 0.466 |
| **EAT-all**[SSL] | 0.709 | 0.704 | 0.448 | 0.315 | 0.694 | 0.166 | 0.677 | 0.348 | 0.611 | 0.788 | 0.512 |
| **sl-BEATS-bio**[SL-SSL] | **0.840** | 0.811 | 0.594 | 0.390 | 0.719 | 0.288 | 0.726 | 0.484 | 0.681 | 0.789 | 0.516 |
| **sl-BEATS-all**[SL-SSL] | 0.832 | **0.813** | **0.604** | **0.408** | **0.726** | **0.294** | **0.732** | 0.511 | 0.690 | 0.798 | 0.529 |
| **sl-EAT-bio**[SL-SSL] | 0.797 | 0.792 | 0.562 | 0.353 | 0.687 | 0.249 | 0.705 | 0.495 | 0.672 | 0.806 | 0.565 |
| **sl-EAT-all**[SL-SSL] | 0.788 | 0.791 | 0.536 | 0.356 | 0.704 | 0.255 | 0.706 | 0.456 | 0.637 | 0.798 | 0.530 |

Table 3: Aggregate results across bioacoustic benchmarks and tasks (best per metric in bold). We report ROC AUC for retrieval, accuracy for probing on BEANS classification and Individual ID, mean-average precision for probe on BEANS Detection and BirdSet. We report the mean of each metric over datasets per benchmark. †BirdNet results on BirdSet are excluded following the authors (Rauch et al., 2025b) due to data leakageModel labels carry training tags: [SSL] self-supervised, [SL] supervised, [SL-SSL] supervised fine-tuning after SSL pretraining. Models above the midrule are existing/pretrained checkpoints; below are new models from this work. EfficientNet models are shaded.

gains of the pre-trained backbone, yielding models which are strong both in and out of distribution, maintaining the benefits of both paradigms - we visualize this in Figure 2b. While post-training both EAT and BEATs gave consistent improvements vs. their raw SSL models (Figure 3 in the Appendix), solely the post-trained BEATs achieved SOTA results overall, possibly suggesting that stronger SSL backbones may lead to better post-trained models. The other existing bioacoustic post-trained self-supervised backbone NatureBEATs follows closely on several benchmarks, significantly outperforming pre-trained BEATs and outperforming supervised baselines on multiple benchmarks. Interestingly, we observe the pre-trained NatureBEATs extends the line of the self-supervised models, while our post-trained models behave more like stronger supervised models (Figure 2b.) This discovery provides interesting signal for future work on post-training under different paradigms. Overall, these results strongly support our proposed recipe of self-supervised training on diverse data-mixes of bioacoustics and general audio, followed by supervised post-training on the same mix. They also show supervised and self-supervised learning in bioacoustics are complementary for representation learning, and suggest a simple step to improve overall quality of self-supervised bioacoustic encoders, not yet commonly adopted (Hagiwara, 2023; Rauch et al., 2025a). We share full, non-aggregated results for all models in the Appendix in Tables 6, 7, 8 and 9.

## 5 CONCLUSION

We presented the first large-scale empirical study and a recipe for developing a generalizable bioacoustic encoder. With few architectural assumptions, we believe this recipe can scale as both labeled bioacoustic data continue to grow and self-supervised learning continues to improve.

We benchmarked 19 CNN- and Transformer-based models across 26 datasets and four task families. We demonstrated that self-supervised pre-training on a mixture of broad bioacoustic and general-audio data, followed by supervised post-training on the same mix, yields the best in- and out-of-distribution results, outperforming state-of-the-art baselines such as BirdNET, Perch and BEATs on

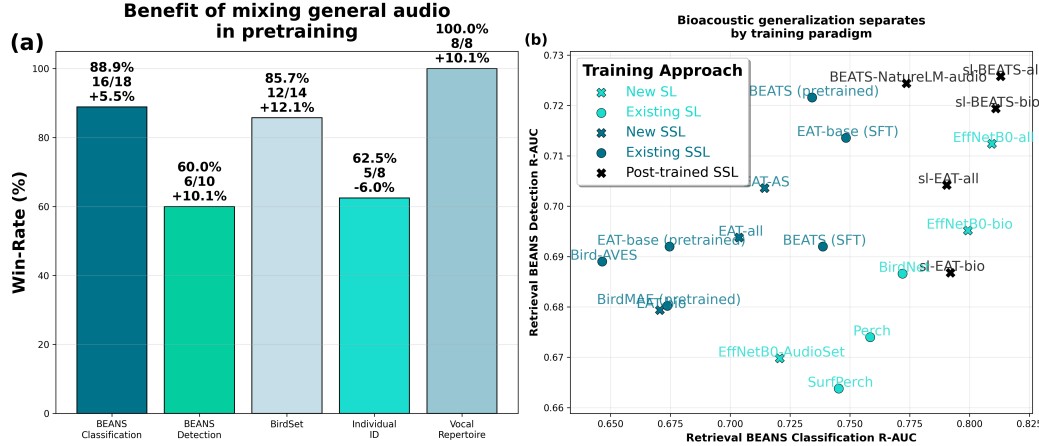

Figure 2: (a) Win-rate of adding AudioSet in self-supervised pre-training vs. pure bioacoustic data, with average relative gain per metric. (b) Supervised encoders outperform self-supervised on BEANS classification, which is primarily focal recordings. However, self-supervised encoders suffer markedly smaller performance drops than supervised encoders when moving from focal recordings to soundscape (BEANS Detection), showing strong out-of-distribution performance. In contrast, self-supervised encoders post-trained with supervised learning on bioacoustic data enjoy the strongest performance both in and out-of distribution.

existing benchmarks (BEANS, BirdSet), and also on newly introduced benchmarks for individual identification and vocal repertoire. Diverse training audio, especially adding AudioSet, consistently improved transfer, whereas supervised training on general audio alone transferred poorly.

Beyond models, we broaden bioacoustic evaluation by curating new benchmarks for individual identification and vocal repertoire classification from public datasets, and by augmenting existing suites with retrieval and clustering metrics. These additions probe representation quality directly and are aligned with practical tasks such as audio-to-audio retrieval and repertoire discovery. We observe that large-scale bioacoustic pre-training is an effective path toward representations that generalize to these under-studied tasks.

Together, these findings provide actionable recipes for building versatile encoders and a richer benchmark for future research. By open-sourcing our encoders[2], we hope this line of work will be used to accelerate research in animal communication and conservation applications through bioacoustics.

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

# A TRAINING AND EVALUATION DATA COMPARISON

To complement the related works discussion in Section 2, we provide a comparison with current bioacoustics benchmarks in Table 4, including both training and evaluation configurations.

Table 4: Training and evaluation data comparison for papers comparing audio encoders across multiple species benchmarks. The ones from Robinson et al. (2025) were reframed for zero-shot learning.

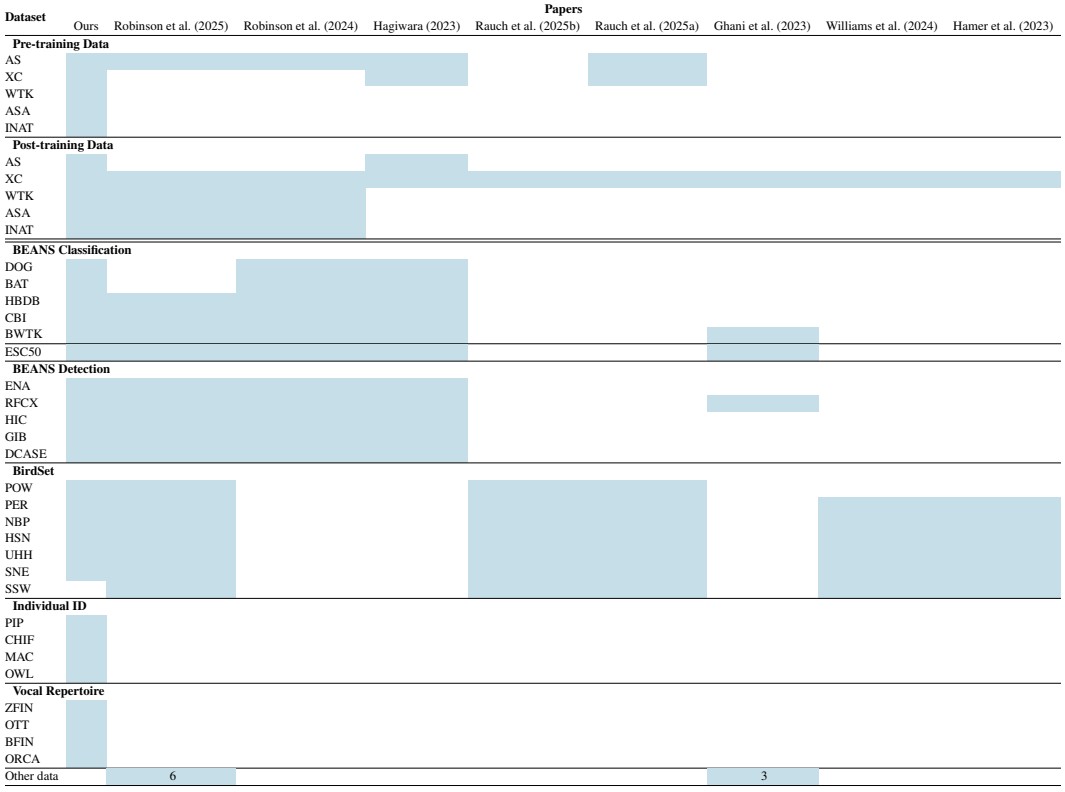

# B EXPERIMENTAL SETUP

## B.1 EVALUATION METRICS

## B.2 PERFORMANCE METRICS

We formalize the evaluation metrics we introduce in Section 3.4. We evaluate linear probing with accuracy for classification, and macro-averaged mean-average precision for detection. We evaluate retrieval with ROC AUC and clustering with NMI. We formalize all evaluation metrics below.

- **Linear Probing Performance**

    1a. Top-1 Accuracy (for classification):

    $$A = \frac{1}{N} \sum_{i=1}^{N} \mathbb{I}(y_i = \hat{y}_i) \tag{1}$$

    where $N$ is the number of samples, $y_i$ is the true label, $\hat{y}_i$ is the predicted label, and $\mathbb{I}$ is the indicator function (Hagiwara, 2023; Rauch et al., 2025b).

1b. **Average Precision (AP).** For each class $k$, let $\pi_k$ be the permutation of $\{1, \ldots, N\}$ that sorts examples by decreasing score for class $k$. Define

$$t_i := \mathbf{1}\{y_{\pi_k(i),k} = 1\}, \qquad TP_i := \sum_{j=1}^{i} t_j, \qquad P(i) := \frac{TP_i}{i}.$$

The (non-interpolated) average precision for class $k$ is

$$\mathrm{AP}_k = \frac{1}{\max\left(1, \sum_{n=1}^{N} y_{n,k}\right)} \sum_{i=1}^{N} P(i)\, t_i,$$

and the mean average precision is the macro average over classses

$$\mathrm{mAP} = \frac{1}{K} \sum_{k=1}^{K} \mathrm{AP}_k \, .$$

- **Retrieval Performance (Area Under the ROC Curve)**:

$$\mathrm{AUC} = \int_0^1 \mathrm{TPR(FPR)} \, d\mathrm{FPR} \tag{2}$$

where $\mathrm{TPR} = \frac{\mathrm{TP}}{\mathrm{TP+FN}}$ and $\mathrm{FPR} = \frac{\mathrm{FP}}{\mathrm{FP+TN}}$, with TP, FP, TN, FN being true/false positives/negatives (Rauch et al., 2025b; Hamer et al., 2023).

- **Clustering Performance (Normalized Mutual Information)**:

$$\mathrm{NMI}(U, V) = \frac{2 \cdot I(U, V)}{H(U) + H(V)} \tag{3}$$

where $U$ and $V$ are cluster assignments, $I(U, V)$ is their mutual information, and $H(U)$, $H(V)$ are their entropies (Kather et al., 2025).

## B.3 WIN-RATE

We quantify the benefit of post-training self-supervised learning (SSL) backbones through a win-rate analysis that compares post-trained models against their corresponding base models across multiple benchmarks and evaluation metrics. For each post-training pair $(M_{\mathrm{post}}, M_{\mathrm{base}})$, where $M_{\mathrm{post}}$ denotes a post-trained model and $M_{\mathrm{base}}$ its corresponding base model, we compute the relative percentage improvement for each metric $m$ and dataset $d$ as:

$$\Delta_{m,d}(M_{\mathrm{post}}, M_{\mathrm{base}}) = \frac{S_{m,d}(M_{\mathrm{post}}) - S_{m,d}(M_{\mathrm{base}})}{S_{m,d}(M_{\mathrm{base}})} \times 100\% \tag{4}$$

where $S_{m,d}(M)$ represents the score of model $M$ on metric $m$ for dataset $d$, defined as: $S_{m,d}(M) = A_d(M)$ for linear probing accuracy on classification tasks, $S_{m,d}(M) = \mathrm{mAP}_d(M)$ for mean average precision on detection tasks, $S_{m,d}(M) = \mathrm{AUC}_d(M)$ for retrieval performance, and $S_{m,d}(M) = \mathrm{NMI}_d(M)$ for clustering performance. Cases where $S_{m,d}(M_{\mathrm{base}}) = 0$ are excluded from the analysis to avoid division by zero.

For each benchmark $B$, we define the set of valid metric-dataset combinations $\mathcal{M}_B = \{(m, d) : m \in \mathrm{Metrics}(B), d \in \mathrm{Datasets}(B)\}$, where $\mathrm{Metrics}(B)$ and $\mathrm{Datasets}(B)$ denote the metrics and datasets associated with benchmark $B$, respectively. For a given post-training pair $(M_{\mathrm{post}}, M_{\mathrm{base}})$ and benchmark $B$, we define a binary win indicator $W_{m,d} \in \{0, 1\}$ for each metric-dataset combination $(m, d) \in \mathcal{M}_B$:

$$W_{m,d} = \begin{cases} 1 & \text{if } \Delta_{m,d}(M_{\mathrm{post}}, M_{\mathrm{base}}) > 0 \\ 0 & \text{otherwise} \end{cases} \tag{5}$$

The win-rate $\omega_B$ for benchmark $B$ and post-training pair $(M_{\mathrm{post}}, M_{\mathrm{base}})$ is then computed as the percentage of wins:

$$\omega_B(M_{\mathrm{post}}, M_{\mathrm{base}}) = \frac{1}{|\mathcal{M}_B|} \sum_{(m,d) \in \mathcal{M}_B} W_{m,d} \times 100\% \tag{6}$$

where $|\mathcal{M}_B|$ denotes the cardinality of $\mathcal{M}_B$ (i.e., the total number of valid metric-dataset combinations for benchmark $B$). Note that $W_{m,d}$ is a binary indicator for individual comparisons, while $\omega_B$ is the aggregated win-rate percentage. To obtain an overall assessment of post-training benefits, we aggregate win-rates across all post-training pairs $\mathcal{P} = \{(M_{\text{post}}^{(i)}, M_{\text{base}}^{(i)})\}_{i=1}^N$, where $N$ is the number of post-training pairs. The aggregated win-rate for benchmark $B$ is:

$$\omega_B^{\text{agg}} = \frac{\sum_{i=1}^N \sum_{(m,d)\in\mathcal{M}_B} W_{m,d}^{(i)}}{\sum_{i=1}^N |\mathcal{M}_B^{(i)}|} \times 100\% \tag{7}$$

where $W_{m,d}^{(i)}$ denotes the win indicator for pair $i$ and metric-dataset combination $(m,d)$, and $|\mathcal{M}_B^{(i)}|$ is the number of valid combinations for pair $i$ on benchmark $B$. Additionally, we compute the average percentage improvement across all comparisons:

$$\bar{\Delta}_B = \frac{1}{\sum_{i=1}^N |\mathcal{M}_B^{(i)}|} \sum_{i=1}^N \sum_{(m,d)\in\mathcal{M}_B^{(i)}} \Delta_{m,d}^{(i)} \tag{8}$$

where $\Delta_{m,d}^{(i)}$ is the improvement for pair $i$ on metric-dataset combination $(m,d)$. Our analysis considers the following post-training pairs: EAT-AS: (sl-EAT-AS, EAT-all), EAT-bio: (sl-EAT-bio, EAT-all), EAT-all: (sl-EAT-all, EAT-all), BEATS-bio: (sl-BEATS-bio, BEATS (pretrained)), and BEATS-all: (sl-BEATS-all, BEATS (pretrained)), where the notation "sl-" indicates a model that has been post-trained with supervised learning on downstream tasks.

### B.4 DATA SOURCES

For training we use a 2021 version of AudioSet, 2023 versions of Watkins "All cuts" and Animal Sound Archive, June 2023 versions of Xeno-canto, iNaturalist. All the training data was released under Creative Commons licenses on the respective platforms, with the exception of Watkins for which we received appropriate licensing agreements, including permission to redistribute. We downloaded BirdSet using the Huggingface dataset library[3]. For BEANS we used the respective scripts the authors provide in their repository[4].

For the new benchmarks we use the public repositories of Bengalese Finch[5], Zebra Finch[6], Giant Otters[7], DCLDE 2026 Killer Whalehttps://catalog.data.gov/dataset/dclde-2026-killer-whale-orcinus-orca-ecotype-and-other-species-annotations-for-the-detecti-2026, Bird ID[8], Macaques Coo Calls[9].

### B.5 SOFTWARE IMPLEMENTATION

Our experimental pipeline is implemented in Python using the Pytorch library. We have used a fixed random seed (42) for generating the datasets and as initial seeds for Pytorch and numpy.

We used open-source implementations for: BEATs[10], EAT[11], and EfficientNetB0 from torchvision[12]. We wrote pytorch wrappers for BirdNet and Perch using tensorflow-lite.

### B.6 HYPERPARAMETERS

We include the full hyperparameters for our trained models in Table 5.

---

[3]https://huggingface.co/datasets/DBD-research-group/BirdSet

[4]https://github.com/earthspecies/beans

[5]https://figshare.com/articles/dataset/Bengalese_Finch_song_repository/4805749

[6]https://www.nature.com/articles/s41467-018-06394-9

[7]https://archive.org/details/giant_otters

[8]https://zenodo.org/records/1413495

[9]https://archive.org/details/macaque_coo_calls

[10]https://github.com/microsoft/unilm/tree/master/beats

[11]http://github.com/cwx-worst-one/EAT

[12]https://docs.pytorch.org/vision/main/models/generated/torchvision.models.efficientnet_b0.html

Table 5: Training hyperparameters for all model variants

| Model | Training Data | Stage-1 Epochs | Stage-1 LR | Epochs | LR | Batch Size | Optimizer | Weight Decay | Scheduler | Warmup Steps |
|---|---|---|---|---|---|---|---|---|---|---|
| *EfficientNet Variants* | | | | | | | | | | |
| EffNetB0-AS | AudioSet | NA | NA | 50 | 5e-4 | 256 | AdamW | 0.01 | Cosine | 4000 |
| EffNetB0-bio | bio | NA | NA | 50 | 5e-4 | 256 | AdamW | 0.01 | Cosine | 4000 |
| EffNetB0-all | AS + Bio | NA | NA | 50 | 5e-4 | 256 | AdamW | 0.01 | Cosine | 4000 |
| EffNetB0-soundscape | Bio + Soundscape | NA | NA | 50 | 5e-4 | 128 | AdamW | 0.01 | Cosine | 4000 |
| EffNetB0-nobirds[‡] | Bio(no birds) | NA | NA | 50 | 5e-4 | 256 | AdamW | 0.01 | Cosine | 4000 |
| EffNetB0-nowhales[‡] | Bio + AS (no whales) | NA | NA | 50 | 5e-4 | 256 | AdamW | 0.01 | Cosine | 4000 |
| EffNetB0-birds[‡] | Bio + AS (birds only) | NA | NA | 50 | 5e-4 | 256 | AdamW | 0.01 | Cosine | 4000 |
| *BEATs Variants* | | | | | | | | | | |
| sl-BEATs-all | All datasets | 2 | 5e-4 | 10 | 1e-4 | 256 | AdamW | 0.01 | Cosine | 5000 |
| sl-BEATs-bio | Bio | 2 | 5e-4 | 10 | 1e-4 | 256 | AdamW | 0.01 | Cosine | 5000 |
| *EAT Variants* | | | | | | | | | | |
| EAT-bio | Bio | NA | NA | 30 | 1e-4 | 48 | AdamW | 0.01 | Cosine | 53333 |
| EAT-all | AS + Bio | NA | NA | 30 | 1e-4 | 48 | AdamW | 0.01 | Cosine | 53333 |
| EAT-AS | AudioSet | NA | NA | 30 | 1e-4 | 48 | AdamW | 0.01 | Cosine | 53333 |
| sl-EAT-bio | SSL + Bio | 2 | 1e-4 | 10 | 8e-5 | 256 | AdamW | 0.01 | Cosine | 2000 |
| sl-EAT-all | SSL + All | 2 | 1e-4 | 10 | 8e-5 | 256 | AdamW | 0.01 | Cosine | 2000 |

[‡] Ablation studies with filtered training data.
BEATs and EAT sl_ models have an initial stage with backbone frozen, with cosine scheduler for stage-1 epochs with Stage-1 lr
Bio = core bioacoustic data, AS = AudioSet, SSL = Self-supervised learning.

To select the hyperparameters we started from the ones proposed in their original papers. In the case of BEATs the learning rate we started from 1e-4 which is the original peak learning rate. We did 5k warmup steps, the same as the original paper. We found that several works, particularly DCASE challenges are doing the same. For EAT we found it important to decrease the learning rate with respect to the original paper (5e-4) because of the larger batch size. For AVES we keep the learning rate (1e-4) which was used in the BEANS benchmark, the test-bed for this model. For probing we use a learning rate of 1e-4, weight decay of 0.1, batch size of 32, and 900 epochs.

# C  ADDITIONAL RESULTS

## C.1  BENEFITS OF POST-TRAINING VS. RAW SSL

As shown in Figure 3, post-training SSL encoders with supervised learning provides a consistent improvement vs. raw SSL backbones, sometimes with large relative gains. These results show that supervised learning can have complementary benefits to self-supervised learning for bioacoustic representation learning. They also give clear evidence for those developing self-supervised models to post-train supervised learning, even when the objective is transfer to out-of-distribution tasks.

## C.2  ABLATION ON TRANSFER OF TRAINING DATA TO DOWNSTREAM TASKS

We show additional ablations on transfer of various data-mixes to downstream tasks in Figure 4 and Figure 5. From a baseline of (focal) bioacoustic data only, we show the performance of adding general audio, adding soundscape recordings, and ablating different taxonomic groups (whales, and then all taxa but birds.) Adding general audio to the training mix improved results overall, but in particular transferred consistently across our vocal repertoire datasets. This data mix yields large gains on the ESC-50 dataset evaluating representations of general audio; though unsurprising, this is a relevant benefit for bioacoustic encoders in tasks such as classifying environmental noise. Training on only general audio data dropped performance very significantly overall, but the drops were most severe on BEANS Classification tasks well-informed by species prediction, and relatively smaller on detection. Adding soundscape data into the training mix with focal is a tempting strategy for learning

**Benefit of Post-training SSL Backbones**

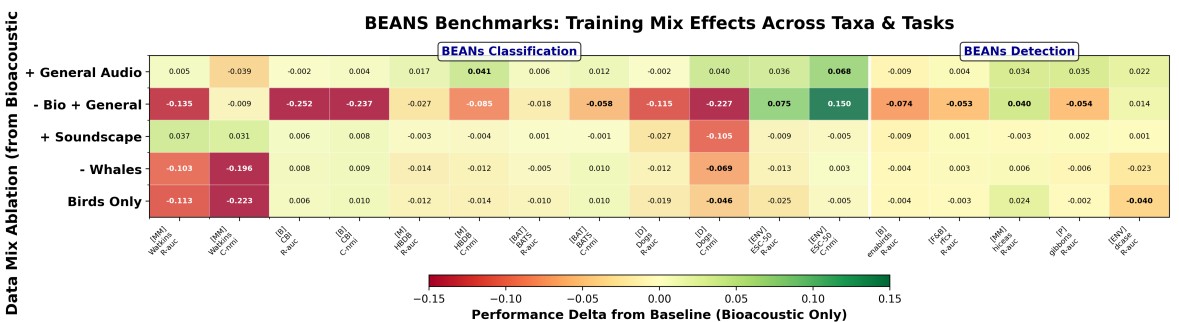

Figure 3: Win-rate of post-trained SSL models vs. their raw SSL backbones. We plot the win-rates summing over all metrics for all our post-trained (EAT and BEATs) models, and show the average relative gain per model with respect to its base model.

**BEANS Benchmarks: Training Mix Effects Across Taxa & Tasks**

Figure 4: Detailed transfer of training data to taxa and tasks in the BEANS benchmark. Heatmap shows the performance change for an EfficientNet trained on each data mix as compared to a baseline "bio" dataset. "- Bio + General" is trained on only AudioSet, "+ Soundscape" adds soundscape datasets, "- Whales" ablates all marine mammal recordings, "Birds only" removes all non-bird recordings.

improved representations useful for downstream tasks on soundscapes, used e.g. by later versions of BirdNet (Kahl et al., 2021). However, in our ablation, this did not give consistent improvements, possibly due to the lack of diversity in the easily accessible soundscape data.

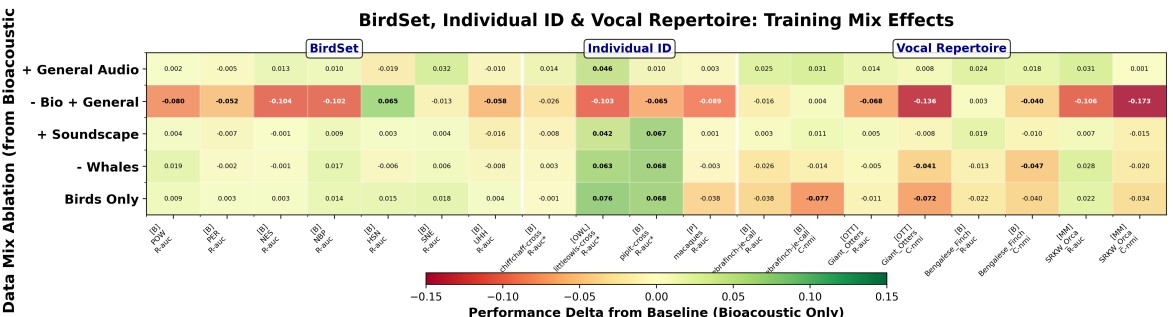

Figure 5: Detailed transfer of training data to taxa and tasks in the BirdSet, Individual Identification, and Vocal Repertoire Discovery benchmarks. Heatmap shows the performance change for an EfficientNet trained on each data mix as compared to a baseline "bio" dataset. "- Bio + General" is trained on only AudioSet, "+ Soundscape" adds soundscape datasets, "- Whales" ablates all marine mammal recordings, "Birds only" removes all non-bird recordings.

Table 6: BEANS Classification datasets only (best per metric in bold). We report R-AUC for retrieval; probe accuracy; clustering reported as NMI. Models above the midrule are existing/pretrained checkpoints; below are new models from this work.

| Model | Watkins Probe | R-AUC | NMI | CBI Probe | R-AUC | NMI | HBDB Probe | R-AUC | NMI | BATS Probe | R-AUC | NMI | Dogs Probe | R-AUC | NMI | ESC-50 Probe | R-AUC | NMI |
|---|---|---|---|---|---|---|---|---|---|---|---|---|---|---|---|---|---|---|
| BEATS (SFT)[SSL] | 0.820 | 0.775 | 0.610 | 0.332 | 0.710 | 0.567 | 0.769 | 0.702 | 0.391 | 0.639 | 0.614 | 0.184 | 0.842 | 0.647 | 0.350 | 0.945 | 0.984 | 0.921 |
| BEATS (pretrained)[SSL] | 0.903 | 0.806 | 0.694 | 0.359 | 0.679 | 0.564 | 0.810 | 0.702 | **0.564** | 0.705 | 0.635 | 0.191 | 0.935 | 0.666 | 0.427 | 0.930 | 0.917 | 0.813 |
| EAT-base (pretrained)[SSL] | 0.850 | 0.744 | 0.585 | 0.247 | 0.617 | 0.502 | 0.778 | 0.630 | 0.482 | 0.635 | 0.588 | 0.125 | 0.705 | 0.585 | 0.194 | 0.858 | 0.884 | 0.655 |
| EAT-base (SFT)[SL-SSL] | 0.867 | 0.808 | 0.613 | 0.388 | 0.714 | 0.558 | 0.782 | 0.686 | 0.328 | 0.654 | 0.631 | 0.216 | 0.899 | 0.659 | 0.216 | **0.958** | **0.992** | **0.938** |
| Bird-AVES-biox-base[SSL] | 0.852 | 0.703 | 0.556 | 0.318 | 0.613 | 0.521 | 0.769 | 0.594 | 0.435 | 0.662 | 0.593 | 0.091 | 0.770 | 0.585 | 0.233 | 0.858 | 0.791 | 0.624 |
| NatureBEATs[SL-SSL] | 0.926 | 0.872 | 0.761 | 0.580 | 0.756 | 0.586 | 0.804 | 0.731 | 0.503 | 0.720 | 0.648 | **0.274** | 0.885 | 0.684 | 0.436 | 0.912 | 0.951 | 0.798 |
| Bird-MAE-Huge[SSL] | 0.888 | 0.744 | 0.567 | 0.457 | 0.623 | 0.537 | **0.829** | 0.695 | 0.470 | **0.733** | 0.580 | 0.083 | 0.827 | 0.577 | 0.244 | 0.860 | 0.823 | 0.691 |
| SurfPerch[SL] | 0.841 | 0.787 | 0.581 | 0.570 | 0.798 | 0.687 | 0.756 | 0.687 | 0.437 | 0.622 | 0.615 | 0.168 | 0.878 | 0.664 | 0.309 | 0.890 | 0.921 | 0.777 |
| BirdNet[SL] | 0.897 | 0.826 | 0.616 | 0.702 | 0.835 | 0.661 | 0.782 | **0.734** | 0.488 | 0.706 | 0.655 | 0.225 | 0.885 | 0.704 | 0.490 | 0.805 | 0.878 | 0.660 |
| Perch[SL] | 0.831 | 0.780 | 0.565 | 0.792 | 0.868 | 0.669 | 0.628 | 0.611 | 0.187 | 0.605 | 0.627 | 0.185 | 0.928 | 0.758 | 0.556 | 0.823 | 0.907 | 0.703 |
| EffNetB0-AudioSet[SL] | 0.708 | 0.759 | 0.753 | 0.235 | 0.660 | 0.531 | 0.732 | 0.666 | 0.310 | 0.566 | 0.621 | 0.156 | 0.799 | 0.649 | 0.312 | 0.868 | 0.969 | 0.852 |
| EffNetB0-bio[SL] | 0.906 | 0.894 | 0.762 | 0.780 | 0.912 | 0.768 | 0.752 | 0.693 | 0.395 | 0.633 | 0.639 | 0.214 | 0.921 | **0.764** | 0.539 | 0.723 | 0.894 | 0.702 |
| EffNetB0-all[SL] | 0.900 | 0.899 | 0.723 | 0.772 | 0.910 | 0.772 | 0.750 | 0.710 | 0.436 | 0.649 | 0.645 | 0.226 | 0.899 | 0.762 | **0.579** | 0.830 | 0.930 | 0.770 |
| EAT-AS[SSL] | 0.855 | 0.802 | 0.640 | 0.266 | 0.633 | 0.520 | 0.800 | 0.718 | 0.489 | 0.654 | 0.632 | 0.212 | 0.784 | 0.604 | 0.236 | 0.868 | 0.897 | 0.743 |
| EAT-bio[SSL] | 0.823 | 0.732 | 0.574 | 0.330 | 0.629 | 0.514 | 0.758 | 0.701 | 0.455 | 0.639 | 0.596 | 0.151 | 0.863 | 0.583 | 0.196 | 0.740 | 0.782 | 0.568 |
| EAT-all[SSL] | 0.873 | 0.773 | 0.618 | 0.326 | 0.644 | 0.516 | 0.791 | 0.722 | 0.475 | 0.655 | 0.612 | 0.162 | 0.755 | 0.593 | 0.227 | 0.853 | 0.878 | 0.689 |
| sl-BEATS-bio[SL-SSL] | **0.935** | 0.911 | 0.786 | 0.798 | 0.933 | 0.801 | 0.775 | 0.702 | 0.470 | 0.696 | **0.656** | 0.205 | **0.942** | 0.730 | 0.499 | 0.897 | 0.934 | 0.805 |
| sl-BEATS-all[SL-SSL] | 0.914 | 0.896 | 0.781 | 0.789 | 0.931 | 0.788 | 0.789 | 0.718 | 0.488 | 0.681 | 0.654 | 0.218 | 0.906 | 0.730 | 0.499 | 0.912 | 0.949 | 0.849 |
| sl-EAT-bio[SL-SSL] | 0.903 | **0.945** | **0.840** | **0.818** | 0.941 | **0.829** | 0.754 | 0.685 | 0.407 | 0.657 | 0.626 | 0.170 | 0.871 | 0.690 | 0.407 | 0.778 | 0.865 | 0.720 |
| sl-EAT-all[SL-SSL] | 0.885 | 0.932 | 0.761 | 0.755 | **0.943** | 0.802 | 0.754 | 0.657 | 0.340 | 0.650 | 0.635 | 0.183 | 0.863 | 0.681 | 0.384 | 0.818 | 0.895 | 0.747 |

Table 7: BEANS Detection datasets only (best per metric in bold). We report R-AUC for retrieval and mean-average precision for probe. Models above the midrule are existing/pretrained checkpoints; below are new models from this work.

| Model | enabirds Probe | R-AUC | rfcx Probe | R-AUC | hiceas Probe | R-AUC | gibbons Probe | R-AUC | dcase Probe | R-AUC |
|---|---|---|---|---|---|---|---|---|---|---|
| BEATS (SFT)[SSL] | 0.428 | 0.643 | 0.094 | 0.713 | 0.577 | 0.584 | 0.216 | 0.673 | 0.381 | 0.847 |
| BEATS (pretrained)[SSL] | 0.525 | 0.678 | 0.110 | 0.720 | 0.544 | **0.627** | 0.351 | 0.686 | 0.373 | 0.897 |
| EAT-base (pretrained)[SSL] | 0.403 | 0.631 | 0.077 | 0.706 | 0.475 | 0.564 | 0.041 | 0.660 | 0.265 | 0.899 |
| EAT-base (SFT)[SL-SSL] | 0.467 | 0.672 | 0.106 | 0.709 | 0.541 | 0.584 | 0.247 | 0.699 | 0.430 | 0.904 |
| Bird-AVES-biox-base[SSL] | 0.465 | 0.646 | 0.111 | 0.711 | 0.472 | 0.612 | 0.344 | 0.626 | 0.309 | 0.850 |
| NatureBEATs[SL-SSL] | 0.601 | 0.714 | 0.124 | 0.764 | **0.596** | 0.624 | 0.159 | 0.627 | 0.447 | 0.893 |
| Bird-MAE-Huge[SSL] | 0.572 | 0.656 | 0.116 | 0.690 | 0.496 | 0.545 | 0.219 | 0.626 | 0.367 | 0.884 |
| SurfPerch[SL] | 0.465 | 0.598 | 0.131 | 0.714 | 0.443 | 0.595 | 0.083 | 0.609 | 0.383 | 0.803 |
| BirdNet[SL] | **0.648** | **0.743** | 0.148 | 0.747 | 0.431 | 0.532 | 0.279 | 0.584 | 0.455 | 0.827 |
| Perch[SL] | 0.610 | 0.643 | **0.149** | **0.783** | 0.464 | 0.530 | 0.252 | 0.622 | 0.365 | 0.792 |
| EffNetB0-AudioSet[SL] | 0.343 | 0.627 | 0.060 | 0.679 | 0.398 | 0.561 | 0.145 | 0.589 | 0.285 | 0.893 |
| EffNetB0-bio[SL] | 0.501 | 0.701 | 0.120 | 0.732 | 0.486 | 0.521 | 0.258 | 0.643 | 0.459 | 0.879 |
| EffNetB0-all[SL] | 0.528 | 0.692 | 0.129 | 0.736 | 0.505 | 0.555 | 0.166 | 0.678 | **0.482** | 0.901 |
| EAT-AS[SSL] | 0.418 | 0.654 | 0.086 | 0.717 | 0.534 | 0.579 | 0.255 | 0.665 | 0.263 | 0.903 |
| EAT-bio[SSL] | 0.428 | 0.660 | 0.087 | 0.665 | 0.571 | 0.515 | 0.081 | 0.667 | 0.389 | 0.890 |
| EAT-all[SSL] | 0.475 | 0.668 | 0.103 | 0.723 | 0.569 | 0.511 | 0.155 | 0.666 | 0.275 | 0.901 |
| sl-BEATS-bio[SL-SSL] | 0.555 | 0.712 | 0.109 | 0.750 | 0.536 | 0.571 | 0.303 | 0.667 | 0.448 | 0.897 |
| sl-BEATS-all[SL-SSL] | 0.566 | 0.716 | 0.118 | 0.741 | 0.527 | 0.566 | **0.366** | **0.700** | 0.465 | **0.906** |
| sl-EAT-bio[SL-SSL] | 0.516 | 0.666 | 0.099 | 0.708 | 0.546 | 0.580 | 0.190 | 0.638 | 0.415 | 0.842 |
| sl-EAT-all[SL-SSL] | 0.528 | 0.665 | 0.099 | 0.739 | 0.536 | 0.618 | 0.170 | 0.667 | 0.445 | 0.832 |

## C.3 FULL RESULTS

We include full results for each benchmark in Tables 6 (BEANs Classification) 7 (BEANS Detection) 8 (BirdSet) and Table 9 (Vocal Repertoire and Individual ID).

Table 8: BirdSet benchmark results: Multi-label bird detection tasks (best per metric in bold). We report ROC AUC for retrieval as R-AUC and mean-average precision for probe. No clustering metrics are reported. [†]BirdNet results are excluded following the authors (Rauch et al., 2025b). Models above the midrule are existing/pretrained checkpoints; below are new models from this work.

| Model | POW Probe | R-AUC | PER Probe | R-AUC | NES Probe | R-AUC | NBP Probe | R-AUC | HSN Probe | R-AUC | SNE Probe | R-AUC | UHH Probe | R-AUC |
|---|---|---|---|---|---|---|---|---|---|---|---|---|---|---|
| BEATS (SFT)[SSL] | 0.108 | 0.654 | 0.046 | 0.642 | 0.062 | 0.726 | 0.213 | 0.649 | 0.107 | 0.625 | 0.079 | 0.725 | 0.094 | 0.704 |
| BEATS (pretrained)[SSL] | 0.157 | 0.703 | 0.070 | 0.649 | 0.095 | 0.731 | 0.248 | 0.648 | 0.105 | 0.568 | 0.116 | 0.753 | 0.109 | 0.751 |
| EAT-base (pretrained)[SSL] | 0.137 | 0.658 | 0.053 | 0.621 | 0.064 | 0.712 | 0.188 | 0.627 | 0.094 | 0.548 | 0.092 | 0.679 | 0.098 | 0.706 |
| EAT-base (SFT)[SL-SSL] | 0.163 | 0.649 | 0.066 | 0.634 | 0.097 | 0.745 | 0.290 | 0.651 | 0.140 | 0.585 | 0.124 | 0.741 | 0.124 | 0.724 |
| Bird-AVES-biox-base[SSL] | 0.142 | 0.679 | 0.044 | 0.615 | 0.050 | 0.755 | 0.196 | 0.631 | 0.050 | 0.556 | 0.082 | 0.714 | 0.081 | 0.740 |
| NatureBEATs[SL-SSL] | 0.244 | **0.722** | 0.132 | **0.690** | 0.177 | 0.819 | 0.419 | 0.708 | 0.251 | 0.574 | 0.197 | **0.796** | 0.143 | 0.749 |
| Bird-MAE-Huge[SSL] | 0.243 | 0.718 | 0.092 | 0.621 | 0.148 | 0.686 | 0.314 | 0.599 | 0.104 | 0.527 | 0.132 | 0.618 | 0.141 | 0.686 |
| SurfPerch[SL] | 0.186 | 0.691 | 0.067 | 0.619 | 0.151 | 0.811 | 0.252 | 0.639 | 0.183 | 0.582 | 0.120 | 0.747 | 0.164 | 0.766 |
| BirdNet[SL] | N/A | N/A | N/A | N/A | N/A | N/A | N/A | N/A | N/A | N/A | N/A | N/A | N/A | N/A |
| Perch[SL] | 0.236 | 0.686 | 0.132 | 0.626 | **0.341** | 0.803 | 0.374 | 0.595 | 0.183 | 0.512 | 0.160 | 0.658 | 0.203 | 0.713 |
| EffNetB0-AudioSet[SL] | 0.115 | 0.637 | 0.045 | 0.569 | 0.054 | 0.728 | 0.181 | 0.615 | 0.087 | 0.609 | 0.087 | 0.725 | 0.115 | 0.701 |
| EffNetB0-bio[SL] | 0.283 | 0.717 | 0.128 | 0.621 | 0.263 | 0.832 | 0.454 | 0.717 | 0.383 | 0.544 | 0.212 | 0.738 | **0.231** | 0.759 |
| EffNetB0-all[SL] | 0.276 | 0.719 | 0.137 | 0.616 | 0.273 | **0.845** | 0.473 | 0.727 | 0.375 | 0.525 | 0.196 | 0.770 | 0.220 | 0.749 |
| EAT-AS[SSL] | 0.147 | 0.698 | 0.060 | 0.638 | 0.074 | 0.761 | 0.230 | 0.646 | 0.138 | 0.588 | 0.112 | 0.723 | 0.114 | 0.739 |
| EAT-bio[SSL] | 0.214 | 0.658 | 0.069 | 0.618 | 0.105 | 0.662 | 0.257 | 0.637 | 0.119 | 0.542 | 0.114 | 0.657 | 0.125 | 0.642 |
| EAT-all[SSL] | 0.188 | 0.702 | 0.065 | 0.649 | 0.113 | 0.731 | 0.303 | 0.648 | 0.185 | 0.568 | 0.147 | 0.708 | 0.158 | 0.734 |
| sl-BEATS-bio[SL-SSL] | 0.304 | 0.707 | 0.150 | 0.629 | 0.279 | 0.836 | **0.496** | **0.737** | 0.349 | 0.627 | **0.226** | 0.766 | 0.213 | 0.781 |
| sl-BEATS-all[SL-SSL] | **0.322** | 0.720 | **0.152** | 0.612 | 0.257 | 0.834 | 0.493 | **0.737** | **0.404** | **0.640** | 0.211 | 0.786 | 0.221 | **0.796** |
| sl-EAT-bio[SL-SSL] | 0.274 | 0.670 | 0.143 | 0.596 | 0.224 | 0.813 | 0.436 | 0.713 | 0.283 | 0.636 | 0.190 | 0.760 | 0.191 | 0.748 |
| sl-EAT-all[SL-SSL] | 0.265 | 0.700 | 0.129 | 0.600 | 0.219 | 0.828 | 0.452 | 0.707 | 0.328 | 0.586 | 0.192 | 0.760 | 0.203 | 0.763 |

Table 9: Complex bioacoustic tasks: Individual ID and Vocal Repertoire analysis (best per metric in bold). We report ROC AUC for retrieval as R-AUC. Individual ID probe is accuracy; Vocal Repertoire reports both R-AUC and NMI. Models above the midrule are existing/pretrained checkpoints; below are new models from this work.

| Model | chiffchaff-cross Probe | R-AUC* | littleowls-cross Probe | R-AUC* | pipit-cross Probe | R-AUC* | macaques Probe | R-AUC* | zebrafinch-je-call R-AUC | NMI | Giant_Otters R-AUC | NMI | Bengalese_Finch R-AUC | NMI | SRKW_Orca R-AUC | NMI |
|---|---|---|---|---|---|---|---|---|---|---|---|---|---|---|---|---|
| BEATS (SFT)[SSL] | 0.185 | 0.470 | 0.290 | 0.663 | 0.061 | 0.500 | 0.963 | 0.775 | 0.651 | 0.295 | 0.815 | 0.545 | 0.898 | 0.742 | 0.657 | 0.359 |
| BEATS (pretrained)[SSL] | 0.180 | 0.536 | 0.263 | 0.700 | 0.093 | 0.486 | 0.985 | 0.827 | 0.707 | 0.352 | 0.848 | 0.577 | 0.848 | 0.653 | 0.697 | 0.409 |
| EAT-base (pretrained)[SSL] | 0.205 | 0.544 | 0.317 | 0.676 | 0.058 | 0.469 | 0.872 | 0.804 | 0.684 | 0.231 | 0.788 | 0.503 | 0.974 | 0.871 | 0.626 | 0.265 |
| EAT-base (SFT)[SL-SSL] | 0.245 | 0.511 | 0.391 | 0.714 | 0.054 | 0.456 | 0.981 | 0.848 | **0.742** | 0.341 | 0.855 | **0.591** | 0.984 | 0.820 | 0.687 | 0.357 |
| Bird-AVES-biox-base[SSL] | 0.230 | 0.521 | 0.292 | 0.634 | 0.118 | 0.492 | 0.967 | 0.840 | 0.660 | 0.253 | 0.751 | 0.484 | 0.872 | 0.757 | 0.621 | 0.318 |
| NatureBEATs[SL-SSL] | 0.185 | 0.489 | 0.359 | 0.711 | 0.112 | 0.524 | 0.984 | 0.857 | 0.704 | 0.351 | 0.862 | 0.586 | 0.943 | 0.835 | 0.736 | 0.438 |
| Bird-MAE-Huge[SSL] | 0.195 | 0.503 | 0.361 | 0.706 | 0.104 | 0.497 | 0.956 | 0.841 | 0.678 | 0.278 | **0.998** | 0.519 | 0.917 | 0.811 | 0.653 | 0.331 |
| SurfPerch[SL] | **0.280** | 0.550 | 0.383 | 0.713 | 0.179 | 0.518 | **0.986** | 0.843 | 0.626 | 0.225 | 0.810 | 0.537 | 0.959 | 0.927 | 0.608 | 0.279 |
| BirdNet[SL] | 0.200 | 0.555 | 0.501 | 0.801 | 0.204 | 0.558 | 0.984 | **0.916** | 0.707 | 0.378 | 0.798 | 0.539 | 0.987 | 0.911 | 0.689 | 0.353 |
| Perch[SL] | 0.210 | 0.500 | **0.649** | **0.847** | 0.288 | 0.570 | 0.973 | 0.904 | 0.657 | 0.284 | 0.854 | 0.585 | 0.959 | 0.896 | 0.561 | 0.206 |
| EffNetB0-AudioSet[SL] | 0.225 | 0.506 | 0.290 | 0.627 | 0.109 | 0.492 | 0.966 | 0.823 | 0.701 | 0.354 | 0.760 | 0.438 | 0.966 | 0.863 | 0.611 | 0.270 |
| EffNetB0-bio[SL] | 0.140 | 0.532 | 0.346 | 0.730 | 0.361 | 0.557 | 0.982 | 0.912 | 0.717 | 0.350 | 0.828 | 0.574 | 0.964 | 0.903 | 0.717 | 0.443 |
| EffNetB0-all[SL] | 0.273 | 0.546 | 0.496 | 0.776 | 0.372 | 0.567 | 0.984 | 0.915 | **0.742** | 0.381 | 0.842 | 0.582 | 0.987 | 0.921 | **0.748** | 0.444 |
| EAT-AS[SSL] | 0.165 | 0.544 | 0.251 | 0.643 | 0.073 | 0.473 | 0.957 | 0.848 | 0.707 | 0.301 | 0.833 | 0.566 | 0.977 | 0.885 | 0.688 | 0.379 |
| EAT-bio[SSL] | 0.175 | 0.540 | 0.307 | 0.689 | 0.115 | 0.474 | 0.914 | 0.804 | 0.654 | 0.246 | 0.809 | 0.543 | 0.934 | 0.816 | 0.630 | 0.260 |
| EAT-all[SSL] | 0.200 | 0.547 | 0.152 | 0.575 | 0.109 | 0.487 | 0.929 | 0.836 | 0.709 | 0.333 | 0.820 | 0.549 | 0.977 | 0.847 | 0.646 | 0.321 |
| sl-BEATS-bio[SL-SSL] | 0.235 | 0.558 | 0.339 | 0.722 | 0.390 | 0.572 | 0.972 | 0.873 | 0.700 | 0.369 | 0.840 | 0.572 | 0.880 | 0.675 | 0.735 | 0.448 |
| sl-BEATS-all[SL-SSL] | 0.225 | **0.574** | 0.413 | 0.755 | **0.428** | **0.580** | 0.977 | 0.850 | 0.718 | **0.426** | 0.832 | 0.554 | 0.897 | 0.681 | 0.746 | **0.457** |
| sl-EAT-bio[SL-SSL] | 0.245 | 0.532 | 0.474 | 0.702 | 0.281 | 0.572 | 0.980 | 0.882 | 0.703 | 0.381 | 0.817 | 0.540 | **0.989** | **0.937** | 0.716 | 0.402 |
| sl-EAT-all[SL-SSL] | 0.195 | 0.509 | 0.354 | 0.688 | 0.326 | 0.557 | 0.949 | 0.795 | 0.718 | 0.338 | 0.789 | 0.501 | 0.980 | 0.898 | 0.703 | 0.383 |

## C.4 BEANS CLASSIFICATION - DATA MIX WITH/WITHOUT ESC-50

The BEANS benchmark contains two auxiliary non-bioacoustics datasets ESC-50 and SpeechCommands. We excluded SpeechCommands because this dataset is irrelevant for non-human bioacoustics applications and speech is a well researched area beyond bioacoustics. In contrast, we included ESC-50 because general sound classification is useful for some conservation tasks like habitat classification, poaching monitoring (gunshots, explosions) and this is the reason why we included it in Tables 3. To disentangle the effects of the data mix and answering the question: "does including general sound in training give better representation?" we report the averaged results without ESC-50. We include the original tables side-by-side for comparison.

Figure 6: Supervised encoders outperform self-supervised on BEANS classification, which is primarily focal recordings. However, self-supervised encoders suffer markedly smaller performance drops than supervised encoders when moving from focal recordings to soundscape (BEANS Detection), showing strong out-of-distribution performance. In contrast, self-supervised encoders post-trained with supervised learning on bioacoustic data enjoy the strongest performance both in and out-of distribution.

We note that the ranking per-model has not changed e.g. sl-BEATs models are better than EffNetB0 models. However, there is less of a gap between 'bio' and 'all' setups, with 'bio' being slightly better in some cases such as EffNetB0.

We include a version of Figure 3b without including ESC-50 when aggregating the results. We note that there is a gap between '-all' and '-bio' models on BEANS detection, with the former models having superior R-AUC.

## C.5 BIRDSET DIRECT POST-TRAINING EVALUATION

We take the SL checkpoints we trained for the post-training phase and we directly evaluate them on the BirdSet dataset by considering solely the logits corresponding to the datasets in BirdSet. This evaluation setup is comparable to the LT (large training) setup in the BirdSet paper with the following additions: (1) more training data and (2) fine-tuning a whole model, hence we call it LT+. It contrasts with the DT (dedicated training) setup which we reported initially in Table 8 i.e. linear probing of a model on each BirdSet subset. Notably, in our case DT is done on top of LT+ and it shows degraded performance. Similarly to the BirdSet paper the LT, and aforementioned LT+, have better results than DT.

Why is LT+ better than DT for BirdSet? Both of the setups contain a domain shift and although we add the same augmentations in LT+ and DT, DT seems to overfit to often small training set originating in Xeno-Cantto, whereas LT+ sees more data, and learns to discriminate with high granularity a high number of classes by learning the time-frequency priors, useful for the domain shift.

Table 10: Aggregate results for BEANS Classification with and without ESC-50. We report ROC AUC for retrieval, accuracy for probing on BEANS classification. We report the mean of each metric over datasets per benchmark. Model labels carry training tags: [SSL] self-supervised, [SL] supervised, [SL-SSL] supervised fine-tuning after SSL pretraining. Models above the midrule are existing/pretrained checkpoints; below are new models from this work.

| Model | BEANS Classification (w/o ESC-50) | | | BEANS Classification (w/ ESC-50) | | |
|---|---|---|---|---|---|---|
| | Probe | R-auc | C-nmi | Probe | R-auc | C-nmi |
| BEATS (SFT)[SSL] | 0.680 | 0.690 | 0.420 | 0.724 | 0.739 | 0.504 |
| BEATS (pretrained)[SSL] | 0.742 | 0.698 | 0.488 | 0.774 | 0.734 | 0.542 |
| EAT-base (pretrained)[SSL] | 0.643 | 0.633 | 0.378 | 0.679 | 0.675 | 0.424 |
| EAT-base (SFT)[SL-SSL] | 0.718 | 0.700 | 0.386 | 0.758 | 0.748 | 0.478 |
| Bird-AVES-biox-base[SSL] | 0.674 | 0.618 | 0.367 | 0.705 | 0.646 | 0.410 |
| NatureBEATs[SL-SSL] | 0.783 | 0.738 | 0.512 | 0.804 | 0.774 | 0.560 |
| SurfPerch[SL] | 0.733 | 0.710 | 0.426 | 0.760 | 0.745 | 0.484 |
| BirdNet[SL] | 0.794 | 0.751 | 0.496 | 0.796 | 0.772 | 0.523 |
| Perch[SL] | 0.757 | 0.729 | 0.432 | 0.768 | 0.759 | 0.478 |
| Bird-MAE-Huge[SSL] | 0.747 | 0.644 | 0.380 | 0.766 | 0.674 | 0.432 |
| EffNetB0-AudioSet[SL] | 0.608 | 0.671 | 0.412 | 0.651 | 0.721 | 0.486 |
| EffNetB0-bio[SL] | 0.798 | 0.780 | 0.536 | 0.786 | 0.799 | 0.563 |
| EffNetB0-all[SL] | 0.794 | 0.785 | 0.547 | 0.800 | 0.809 | 0.584 |
| EAT-AS[SSL] | 0.672 | 0.678 | 0.419 | 0.704 | 0.714 | 0.473 |
| EAT-bio[SSL] | 0.683 | 0.648 | 0.378 | 0.692 | 0.671 | 0.410 |
| EAT-all[SSL] | 0.680 | 0.669 | 0.400 | 0.709 | 0.704 | 0.448 |
| sl-BEATS-bio[SL-SSL] | **0.829** | **0.786** | 0.552 | **0.840** | 0.811 | 0.594 |
| sl-BEATS-all[SL-SSL] | 0.816 | **0.786** | **0.555** | 0.832 | **0.813** | **0.604** |
| sl-EAT-bio[SL-SSL] | 0.801 | 0.777 | 0.531 | 0.797 | 0.792 | 0.562 |
| sl-EAT-all[SL-SSL] | 0.781 | 0.770 | 0.494 | 0.788 | 0.791 | 0.536 |

Table 11: BirdSet benchmark results: Comparison of post-training (LT+) vs adding a dataset-wise probing afterwards, on top of post-training (DT) for sl-BEATS models.

| Model | POW | PER | NES | NBP | HSN | SNE | UHH |
|---|---|---|---|---|---|---|---|
| sl-BEATS-bio (DT) | 0.304 | 0.150 | 0.279 | 0.496 | 0.349 | 0.226 | 0.213 |
| sl-BEATS-bio (LT+) | 0.355 | 0.167 | 0.372 | 0.535 | 0.377 | 0.261 | 0.271 |
| sl-BEATS-all (DT) | 0.322 | 0.152 | 0.257 | 0.493 | 0.404 | 0.211 | 0.221 |
| sl-BEATS-all (LT+) | 0.343 | 0.167 | 0.356 | 0.535 | 0.406 | 0.268 | 0.224 |

## C.6 Probing ablation linear vs attention

In our initial evaluation, the embeddings extracted from the model are averaged on the time axis. To model the temporal dependencies between the embeddings we evaluate some of the models on the BEANS and BirdSet benchmarks using an attention-based probe.

The attention has a single multi-head self-attention layer on top of the extracted backbone representations. The output of this attention operation is added back to the original embeddings through a residual connection, followed by layer normalization and optional dropout. The resulting sequence is then aggregated by taking the mean across tokens, and finally passed through a linear classifier to produce the prediction.

For a fair comparison and to reduce computational cost we train both probe heads for each dataset in BEANS and BirdSET for 50 epochs, instead of the 900 used in our initial experiments. To reduce overfitting, we introduced a cosine learning rate scheduler with the first 5 epochs being the learning stage. We use a learning rate of 0.0001 and an AdamW optimizer. The BirdSet models are trained with the noise and mixup augmentations introduced in Section 3.4.

The results presented in Table 7 show that in general attention probes have superior performance to linear probes. This slightly alters the ranking of the models, e.g. BEATs (pretrained) is now the top model on BEANS Detection, EffNetB0 is surpassed by the transformers, our sl-BEATs-all is still

| | BEANS Classification | | BEANS Detection | | BirdSet | |
|---|---|---|---|---|---|---|
| BEATs (PT) (SSL) | 0.79 | 0.71 | 0.44 | 0.27 | 0.16 | 0.10 |
| NatureBEATs (SL-SSL) | 0.78 | 0.60 | 0.37 | 0.25 | 0.21 | 0.11 |
| EAT-base (PT) (SSL) | 0.77 | 0.69 | 0.38 | 0.32 | 0.15 | 0.12 |
| EAT-base (SFT) (SL-SSL) | 0.76 | 0.47 | 0.35 | 0.22 | 0.12 | 0.07 |
| AVES (SSL) | 0.57 | 0.32 | 0.26 | 0.11 | 0.10 | 0.06 |
| EffNetB0-all (SL) | 0.72 | 0.73 | 0.21 | 0.22 | 0.18 | 0.18 |
| EAT-all (SSL) | 0.77 | 0.51 | 0.39 | 0.23 | 0.16 | 0.11 |
| sl-EAT-all (SL-SSL) | 0.80 | 0.70 | 0.34 | 0.27 | 0.24 | 0.17 |
| sl-BEATs-all (SL-SSL) | 0.80 | 0.80 | 0.40 | 0.38 | 0.26 | 0.25 |
| | Attention | Linear | Attention | Linear | Attention | Linear |

Figure 7: Linear vs Attention probing comparison. The results are aggregated across bioacoustic benchmarks and tasks . We accuracy for BEANs Classification, mean-average precision for BEANs Detection and BirdSet. We report the mean of each metric over datasets per benchmark. Model labels carry training tags: **SSL** self-supervised, **SL** supervised, **SL-SSL** supervised fine-tuning after SSL pretraining. PT denotes pretrained. Models above the red line are existing checkpoints; below are new models from this work.

one of the best models, EAT models improve a lot with the attention head. There are less differences between the top and the bottom models in the ranking.

SSL models benefit more from an attention head. This effect may stem from the training dynamics of SSL models, which emphasize capturing temporal structure in audio rather than developing species-specific inductive biases, as is more common in supervised learning. Consequently, when the backbone is a transformer trained via SSL, pairing it with a transformer-based probe is more effective, as it better aligns with and leverages the backbone's representational properties. Moreover, for EffNetB0-all and sl-BEATs-all the attention probes did not yield considerable gains.

