# OpenReview forum: "AVEX: What Matters for Animal Vocalization Encoding"
_ICLR.cc/2026/Conference — ICLR 2026 Poster_

### Official Review · Reviewer_ykFi · 2025-10-29

**Soundness:** 3
**Presentation:** 2
**Contribution:** 2
**Rating:** 4
**Confidence:** 4

**Summary:**

This paper presents a large-scale empirical study to identify "what matters" for training bioacoustic encoders. It investigates the impact of model architectures, training data composition (bioacoustics-only, general audio, and mixed corpora), and training paradigms (SL, SSL /pre-training and post-training recipes). The authors collect a large-scale pre-training dataset and evaluate on an extended evaluation set. Their key finding is that a recipe of self-supervised pre-training on a mixed bioacoustic and general audio dataset, followed by supervised post-training, yields best performing models in- and out-of-distribution tasks.

**Strengths:**

- **Comprehensive and valuable empirical study**: The primary strength of this work is its ambition and scope. The paper tackles an important question in the bioacoustics community by systematically comparing different modeling components. This kind of large-scale, well-structured empirical study is valuable for future research.
- **Expanded evaluation framework:** The authors curate and introduce a broader evaluation benchmark.  The paper moves beyond species classification and includes tasks like individual ID and vocal repertoire discovery. It also augments existing benchmarks with clustering and retrieval metrics, providing a potentially more realistic assessment of an encoder's performance.
- **Actionable insights:** The paper provides actionable results for training robust bioacoustic encoders. The finding that a combined self-supervised and supervised approach on a diverse data mix yields the best overall performance is an important insight.

**Weaknesses:**

While the findings of the paper are valuable, its contribution is limited by its presentation, methodological choices/novelty, and questions about its reproducibility. A short disclaimer: Given that the paper's contribution is not methodological, the focus necessarily shifts to the quality and execution of its empirical investigation which poses some questions.

**[Presentation and readability]**

- Clarity of figures and tables: The presentation of results could be significantly improved. Figure 2 is difficult to read due to its small size. The tables, particularly Tables 2, 3 and 4, use a complex system of abbreviations that makes them hard to parse and compare models at a glance. A more structured in-table organization with sub-information could help here.
- Lack of structure in text: Long sections of text, such as the Related Work and Proposed Models or Evaluation Setup sections, would benefit greatly from some kind of sub-headers. This would improve readability and help the reader navigate the information being presented.
- Formatting issues: A few minor formatting issues exist, such as Figure 1 being over the text height, placeholders like "ADD REF TO AGILE PAPER" (L.316), and Figure 2 being very hard to read.

**[Methodological Choices]:**

- Dataset and model selection: The paper lacks a clear motivation for the specific choice of datasets in the training corpus and the selection of models for comparison. For example, the rationale for choosing BEATs, for which the official training implementation is unavailable, over a more recent and reproducible open-source model like SSLAM from last year’s ICLR is not provided.
- Incomplete baselines: While the study includes many models, it omits direct comparisons with SSL bioacoustic models mentioned in the related work. Including these would provide a more complete picture and better contextualize the performance of the proposed recipes as the respective baselines.

**[Evaluation details and reproducibility]:**

- Lack of statistical and methodological detail:  It is unclear how the "win-rate" in Figure 2 is calculated or whether any statistical significance testing  (or repetitions) was performed to validate the results. Furthermore, the process for hyperparameter selection across the many models and training recipes is not described, making it difficult to assess the fairness of the comparisons.
- Limited probing techniques: The evaluation relies exclusively on linear probing. While standard, recent work in self-supervised learning in computer vision has shown attentive probing can provide a more nuanced view of an embedding's quality, and its omission here is a missed opportunity. If there were more methodological contributions, this would not be a problem but it would greatly strengthen the contribution of an empiricial study.
- No code or release plan: For a study of this complexity, reproducibility is important, but there is no code available during the review process. The promise to release checkpoints "upon paper acceptance" is not enough. There is also no clear plan for releasing (if it happens) the curated datasets, splits, and evaluation protocol to establish this work properly.

**Questions:**

1. Could you provide a clearer rationale for your selection of training datasets and baseline models? Specifically, why was (the older) BEATs chosen over other reproducible open-source SSL models?
2. Could you clarify some details of the evaluation? (a) How is the "win-rate" metric in Figure 2 formally defined and calculated? (b) Were statistical significance tests (or the mean of some repetitions) performed to validate the superiority of your proposed recipe? (c) Why was only linear probing used and not other methods that might better utilize the learned representations (and change results)?
3. To ensure its impact and allow for verification, can you provide access to the code for the review process? Furthermore, what is your concrete plan for releasing not just the model checkpoints, but (maybe) also the curated data splits or the full evaluation framework to the community?

---

> ### Author Response · Authors · 2025-11-21
>
> We thank the reviewer for useful comments that improve the paper. We found it encouraging that the reviewer valued our “comprehensive and valuable empirical study”, “expanded evaluation framework”, and “actionable insights”. We will address the weaknesses and questions within the paper and we detail this in the following paragraphs.
>
> **Presentation issues**
>
> - We increased the font size in Figure 2\.  Our experiments involving testing multiple combinations of architectures, training paradigms, and datasets require multiple abbreviations. We provide a legend for each checkpoint we trained in Table 2 and we keep these model names in the remaining tables. Captions in the results tables contain enhanced information.
> - We included sub-headers in Sections 2 and 3.4.
> - We have fixed the minor formatting issues.
>
>
> **Methodological choices**
>
> - We give an overview of what training and evaluation data in bioacoustics in Table 4\. We motivate the usage of cross-taxa and general audio data in Section 2 lines 129-139, the extended evaluation with the new benchmarks Section 2 lines 149-163. For models we chose different paradigms SSL and SL and different architectures. We try to cover SotA models currently deployed across many audio downstream tasks.
> - Regarding the choice of BEATs (or EAT which we pre/post-train, or AVES and now BirdMAE which we evaluate). We have included a discussion with respect to SSLAM in the general comments to all the reviewers, point 1\. We have added BirdMAE as the missing SSL bioacoustics model and discussed this in the same point 1\.
>
> **Evaluation**
>
> - We included more details about win-rate computation in the Appendix. To select the hyperparameters we started from the ones proposed in their original paper and also in the related works that fine-tuned BEATs or EAT. In the case of BEATs the learning rate we started from 1e-4 which is the peak learning rate used in the original BEATs. We did 5k warmup steps, the same as the original paper. We found that several works, particularly DCASE challenges are doing the same. For EAT we found it important to decrease the learning rate with respect to the original paper (5e-4) because of the larger batch size. For AVES we keep the learning rate (1e-4) which was used in the BEANS benchmark, the test-bed for this model. For probing we use a learning rate of 1e-4, weight decay of 0.1, batch size of 32, and 900 epochs. We have included this information in the Appendix.
> - We included experiments with linear vs attention-based probe comparison in the Appendix. We added a discussion in the general comments above, point 4\.
> - We addressed this in the general comments, point 7\.
>
>
> **Questions** (it seems a lot of overlap with the Evaluation section)
>
> - We have included a discussion on this issue in the general comments point 1\.
> - We have addressed this point above.
> - We have addressed this point above.

---

> ### Comment · Reviewer_ykFi · 2025-11-26
>
> Thank you for addressing the weaknesses raised in the initial review.
>
> A quick note: In the future, I would strongly recommend to provide the specific changes alongside the responses to each reviewer directly with citations (even if this means copying some points). Having to scroll through the comments to search for the respective answers creates an additional layer of work during the rebuttal process.
>
> Regarding the content, I think the paper is objectively better following your revisions The readability has improved, and several rough edges have been smoothed out. However, I still find it difficult to fully re-evaluate the work due to my concerns regarding experimental rigor and reproducibility. While I understand the constraints on the number of experiments, a study of this nature (without much methodological novelty) requires a higher standard of validation:
>
> - random seeds: I maintain that running experiments with multiple random seeds would improve the statistical validity of the work. Even if the variance is expected to be low, this is standard practice to ensure robustness.
>
> - hyperparameters: While your newly added selection process appears reasonable, the method for determining these hyperparameters lacks rigor. A more systematic approach or justification would improve the methodology.
>
> - code: The absence of code makes it impossible to verify the implementation or reproduce the results, which is a drawback for an empirical study.
>
> So while the paper offers contributions to the community, the lack of scientific rigor prevents me from fully endorsing the results.
>
> A short heads up on score changes: I am currently weighing the improvements in clarity against the issues listed above. I am considering raising my score to a 6, but I have not yet reached a final decision.

---

> > ### Author Response · Authors · 2025-11-26
> >
> > We thank the reviewer for appreciating the improvements and the fact that the paper is objectively better. We would like to re-address the issues mentioned, and we hope that the reviewer values the contributions of the paper beyond these issues which can be addressed in the future.
> >
> > - Random seeds and hyperparameters: First, we would like to remind the reviewer that multiple runs solely change the probe scores and not clustering and retrieval (training-free evaluation). All the findings still hold if we based them solely on these two metrics. The seeds do not control the dataset splits which are fixed. Second, we would really like to run x times more experiments over multiple seeds and perform a grid search for the hyperparameters. Given that the current experiments took a few weeks to run with a considerable cost, this is not feasible to run within one week, until the rebuttal is due, it would likely take 6 weeks for an additional seed. As a reminder we have trained multiple large models and evaluated probing on 20 models over 26 datasets (520 runs with 900 epochs).
> > - We will share privately and anonymously with the reviewer the code at the current state with the disclaimer that we are currently working on making it available so others may use the models and reproduce the experiment. However, at the current state we can offer transparency but not fully reproducibility (although the reviewer may adapt the code to use all the open datasets with their desired method).

---

> > > ### Author Response · Authors · 2025-11-26
> > >
> > > As required by the reviewer we open the code base for reviewing purposes. We are currently working on opening this repository as part of a library and an API that allows to reproduce all the results in the paper but also allows usage on various bioacoustic tasks by the whole community.
> > > https://anonymous.4open.science/r/representation-learning-iclr-0DF9

---

> > > ### Comment · Reviewer_ykFi · 2025-11-27
> > >
> > > Thanks for the code.
> > >
> > > Regarding the hyperparameters and seeds: I understand the computational constraints of the rebuttal period and I wasn't expecting you to do them. I just wanted to point out what still impacts my evaluation of this work. Since the paper positions itself as an empirical investigation, its value is very dependent on the strength and stability of the reported trends. Otherwise it wouldn't have such an impact. In the absence of method-based novelty, the empirical evidence must be indisputable. As stated before, I am weighing this against the improvements.

---

> > > > ### Author Response · Authors · 2025-12-02
> > > >
> > > > We appreciate the fact that the reviewer was willing to change the score. Reviewer WxmL noted that adding various seeds to a probing setup does not account for a large increase in variability of results and findings. After conducting a study of the literature on audio encoding and large models (including BEATs, EAT, Bird-MAE, PaSST, MAEST, HTS-AT, Perch), we found that this very expensive setup is not very common, and these SotA models are usually evaluated for a single seed. All the “sl-” models we consider start from pre-trained checkpoints and random seeds do not control the weights initialization, so there is less variation with respect to that. We believe that this is a valid research question, however the cost and scope of experiments are beyond what we aim at with this paper.

---

### Official Review · Reviewer_WxmL · 2025-10-30

**Soundness:** 3
**Presentation:** 3
**Contribution:** 3
**Rating:** 6
**Confidence:** 4

**Summary:**

The paper investigates the factors that contribute to effective bioacoustic
encoding by evaluating design choices like data sources, model architectures and
training strategies. The authors therefore conduct an empirical study comparing
multiple pre-training datasets (general audio recordings with AudioSet, birds
from Xeno-Canto, Sapsucker Woods and WABAD, diverse animal sounds from
iNaturalist and Animal Sound Archive, marine mammals from Watkins), model
architectures (EfficientNet, ViT) and training strategies (supervised pre- and
post-training, self supervised pretraining with EAT). Evaluation is carried out
on the BEANS and BirdSet benchmark and two additional tasks of individual
identification and vocal repertoire discovery. Generally, models benefit from
diverse pretraining data (also including AudioSet and not only bioacoustic
data). In addition, supervised posttraining improves performance for
SSL-pretrained models. The authors wrap these findings in a recipe for post-training SSL-backbones.

**Strengths:**

- Research topic is relevant and motivated well.
- Contributions are clearly stated.
- Experimental design is comprehensive, covering multiple datasets,
  architectures, and training strategies.
- Results are novel but not entirely surprising.

**Weaknesses:**

- For a systematic comparison of model architectures and training paradigms the study lacks SSL models based on CNN architectures and SL models based on ViTs. This would help to better disentangle the effects of architecture and training paradigm.
- To simplify from the results of one SSL-pretrained model (EAT) to SSL in general is a bit of a stretch. Including more SSL training techniques would strengthen the claims.
- The evaluation protocol only considers the "clip" embeddings and leaves the patch embeddings for future work. [1,2] shows that the general audio models compare better when using patch embeddings with attentive probing.
- BirdMAE as a strong bioacoustic SSL baseline is missing in the comparison.
- The 2 and 3.4 would benefit from paragraph headings to improve readability.
- Glitch in l. 316

[1] Can Masked Autoencoders Also Listen to Birds? https://arxiv.org/abs/2504.12880
[2] Foundation Models for Bioacoustics -- a Comparative Review
https://www.arxiv.org/abs/2508.01277

**Questions:**

- How were the hyperparameters selected? Why are different LR used for BEATs and EAT? What HP are used for linear probing?
- How many seeds were used for the experiments? What are the standard deviations?
- Which BEATs checkpoint was used?
- Do you plan to release the training and evaluation code?
- Analysis of the "curation level" of the datasets would be interesting as initially analyzed in [1]. How many samples per class are needed, how diverse should the data be? Have you considered analyzing this aspect?

---

> ### Author Response · Authors · 2025-11-21
>
> We thank the reviewer for their suggestions and we hope we addressed them all in the paper. Notably, we included Bird-MAE-Huge as another SSL model and updated all tables and figures. The reviewer valued the clarity of our presentation, the comprehensiveness of our evaluation, and the novelty of the results. We address the weakness and the questions in the following paragraphs.
>
> **Weaknesses**
>
> - We have not tried an SSL-based CNN because SSL is not a very common pretraining paradigm for CNNs, the majority of the models are transformers. The EffNets we use are pretrained on ImageNet. We do use ViT but in an SL-SLL paradigm (EAT models). We argue that disentangling the role of architecture is a complex task, involving capacity, size and variety of training data, along-side the training paradigm and maybe should be the subject of a different study.
> - We are post-training other SSL models besides EAT, namely BEATs and AVES. Pre-training is generally a costly paradigm and this is why we chose to train EAT.
> - We included a probing section in Appendix Section C6 and also a discussion in the general comments to all the reviewers, point 4
> - We have included Bird-MAE-Huge in the evaluation, and a discussion in the general comments to all the reviewers, point 1
> - We have added paragraph headings in Sections 2 and 3.4, fixed the glitch
>
>
> **Questions**
>
> - To select the hyperparameters we started from the ones proposed in their original paper and also in the related works that fine-tuned BEATs or EAT. In the case of BEATs the learning rate we started from 1e-4 which is the peak learning rate used in the original BEATs. We did 5k warmup steps, the same as the original paper. We found that several works, particularly DCASE challenges are doing the same. For EAT we found it important to decrease the learning rate with respect to the original paper (5e-4) because of the larger batch size. For AVES we keep the learning rate (1e-4) which was used in the BEANS benchmark, the test-bed for this model. For probing we use a learning rate of 1e-4, weight decay of 0.1, batch size of 32, and 900 epochs. We have included this information in the Appendix.
> - We used the BEATs\_iter3\_plus\_AS2M.pt checkpoint for BEATs (pretrained) and BEATs\_iter3\_plus\_AS2M\_finetuned\_on\_AS2M\_cpt2.p for BEATs (SFT)
> - We addressed this in the general comments to all the reviewers, point 7
> - We did not “data curation” in this paper and we believe it is worth a paper on its own. We are using weakly labeled data in training, rather than fixed size segments, similarly to Perch 2.0. With the taxa ablations and with adding general audio we tried to see whether data mix, and implicitly training data diversity, affects the downstream performance in bioacoustic encoding.

---

> > ### Comment · Reviewer_WxmL · 2025-11-26
> >
> > Dear Authors,
> > Thanks for your response and for adding BirdMAE, paragraph headings and attentive probing results. I also appreciate answering my questions and acknowledge that pretraining is resource intensive.
> >
> > I agree, SSL + CNNs is not very common nowadays, but ViT + SL is. Still, in my humble opinion, to make claims of the influence of the training paradigm only the training paradigm should be different between experiments. The question remains how an $EAT\-all^{SL}$ model would compare, how big is the influence of SSL pretraining? Is it worth it?
> >
> > Formatting: some missing white spaces before citations, e.g. l. 152

---

> > > ### Author Response · Authors · 2025-11-26
> > >
> > > Dear reviewer, thank you for acknowledging our rebuttal and appreciating our answers to your questions.
> > > Regarding the systematic analysis of the training paradigm, we agree that it is an interesting experiment, however we argue that the training paradigm is often tightly coupled with the architecture (see our previous point regarding SSL and CNNs). It also adds another layer of complexity and experiments to an already expensive setup. We think that there are a few reasons to discard a pure SL ViT training (from a randomly initialized checkpoint):
> > >
> > > - It may be considered as a common paradigm but we think mostly for single-task papers (e.g. ViT for emotion recognition \[4\] and even there researchers would not normally start from scratch). Using randomly initialized ViT on audio spectrograms as a foundation model does not seem that common to us, but we would appreciate any SotA reference we may have missed.
> > > - The only SotA papers we found on learning audio representations for downstream tasks using vision transformers on spectrograms in an SL paradigm are PaSST \[1\] and MAEST \[2\]. In the original PaSST paper, section 3.2 “ImageNet Pretraining” says: “Therefore, we will use pre-trained models in all our experiments. Our base model is DeiT-B↑384.” In MAEST they actually compare between pretrained models (DeiT or PaSST) vs Random init and the former have superior performance.
> > > - Pure SL ViTs perform well if trained on large amounts of data (Section 4.3 in \[3\]) which for bioacoustics is one order of magnitude lower than general audio, not to mention the image domain.
> > >
> > > For these reasons, we think that not much gain can be expected by running these additional costly experiments, as the evidence from the literature suggests that it will not give better results. But we will be happy to reconsider this if you have strong reasons to think that it would provide beneficial insights in the specific case of bioacoustics.
> > > We also think that, besides this lacking experiment, this paper has many contributions that we hope you value, and that you’ll continue supporting accepting our paper.
> > >
> > > \[1\] Koutini, K., Schlüter, J., Eghbal-Zadeh, H., & Widmer, G. (2021). Efficient training of audio transformers with patchout. arXiv preprint arXiv:2110.05069.
> > > \[2\] Alonso-Jiménez, P., Serra, X., & Bogdanov, D. (2023). Efficient supervised training of audio transformers for music representation learning. arXiv preprint arXiv:2309.16418.
> > > \[3\] Dosovitskiy, A. (2020). An image is worth 16x16 words: Transformers for image recognition at scale. arXiv preprint arXiv:2010.11929.
> > > \[4\] Akinpelu, S., Viriri, S., & Adegun, A. (2024). An enhanced speech emotion recognition using vision transformer. Scientific Reports, 14(1), 13126\.

---

> > > > ### Comment · Reviewer_WxmL · 2025-11-26
> > > >
> > > > Dear authors,
> > > >
> > > > I agree that it is **probably** not beneficial to train solely with supervision from scratch to achieve SotA performance in bioacoustics. But, as far as I understand, your paper is about answering 'what matters for bioacoustic encoders', in other words, removing the probably. In my humble opinion, your experimental methodology is not answering if SSL pretraining is beneficial as your architectures differs.  So what is against training an $EAT\-all^{SL}$ checkpoint to see how much better the $EAT\-all^{SSL-SL}$ one is? From my point of view, this is necessary to answer your main research question.

---

> > > > > ### Author Response · Authors · 2025-11-26
> > > > >
> > > > > The experiment requires training three large EAT models (-all, -bio, -as) on a large dataset and then evaluating them which is not feasible to complete within less than a week. We can present these new models in addition to the 20 existing ones, as we did with the BirdMAE experiment, however after the rebuttal and before publishing. We believe that they will not change the ranking of the models, the message and the conclusions of the paper.

---

> > > > > > ### Comment · Reviewer_WxmL · 2025-11-28
> > > > > >
> > > > > > Dear authors,
> > > > > >
> > > > > > After considering the revision and the other reviewers arguments, I decided to keep my score as it is.
> > > > > > I agree with the other reviewers that your empirical work needs strong experimental backing.
> > > > > > As stated above, I am missing experiments to quantify the influence of SSL pretraining.
> > > > > > About seeds: While I agree that the linear probing is fairly robust, and you provide plenty of evaluation experiments that will also mitigate random effects, this is not the case for the training of the models. The question remains, if this is neglectable as you stated.
> > > > > >
> > > > > > To keep the computational costs in a reasonable range, I would suggest the following experiments:
> > > > > > - Train $EAT-all^{SL}$ and compare with $EAT-all^{SSL-SL}$.
> > > > > > - Train a second random seed of $EAT-all^{SSL-SL}$ and report the difference in evaluation.

---

> > > > > > > ### Author Response · Authors · 2025-12-02
> > > > > > >
> > > > > > > After conducting a study of the literature on audio encoding and large models (including BEATs, EAT, Bird-MAE, PaSST, MAEST, HTS-AT, Perch) we found that this very expensive multi-seed setup is not very common, and these SotA models are usually evaluated for a single seed.  All the “sl-” models we consider start from pre-trained checkpoints and random seeds do not control the weights initialization, so there is less variation with respect to that. We believe that this is a valid research question, however the cost and scope of experiments are beyond what we aim at with this paper.
> > > > > > >
> > > > > > > We are willing to include a comparison with a pure-SL method within a transformer. However, we argue that starting from a random checkpoint vs a pre-trained one, as we do now, would not significantly change the message of the paper i.e. we would not note better results for the pure SL method. We base this assumption of the current findings in the literature \[1,2\] which found that using a random initialization leads to inferior results. In fact, starting from scratch is something that researchers would do pre-SSL-era when they were developing models for single datasets and not when evaluating representations e.g. emotion recognition \[3\].
> > > > > > >
> > > > > > > \[1\] Koutini, K., Schlüter, J., Eghbal-Zadeh, H., & Widmer, G. (2021). Efficient training of audio transformers with patchout. arXiv preprint arXiv:2110.05069.
> > > > > > > \[2\] Alonso-Jiménez, P., Serra, X., & Bogdanov, D. (2023). Efficient supervised training of audio transformers for music representation learning. arXiv preprint arXiv:2309.16418.
> > > > > > > \[3\] Akinpelu, S., Viriri, S., & Adegun, A. (2024). An enhanced speech emotion recognition using vision transformer. Scientific Reports, 14(1), 13126\.

---

### Official Review · Reviewer_B3CZ · 2025-10-31

**Soundness:** 4
**Presentation:** 3
**Contribution:** 3
**Rating:** 6
**Confidence:** 4

**Summary:**

This paper focuses on building strong bioacoustic embedding models that can be used for a wide range of ecological applications including species ID, individual ID and repertoire discovery. The authors explore supervised and self-supervised methods with a focus on data diversity and scale to suggest a training recipe that works well for bioacoustics tasks, particularly when transferring to novel domains. They benchmark existing methods extensively across 26 datasets, different model architectures, training recipes and data distributions. Part of this benchmarking are the individual-ID and repertoire discovery tasks for which the authors propose newly curated evaluation datasets. Based on their results, the authors recommend training recipes and best practices for learning strong bioacoustic embeddings.

**Strengths:**

1. The paper includes extensive experiments with multiple model architectures, different training recipes and a number of evaluation tasks, both well established and newly proposed.
2. The paper presents a strong benchmark unifying different existing approaches. Different works often follow different evaluation strategies in bioacoustics; and the baselines and evaluation protocol in this paper could serve as a good standardized benchmark and reference for future work.
3. The architecture, pre/post-training recipes and design choices for evaluation tasks are sufficiently detailed and well-motivated.
4. The newly curated tasks for individual ID and repertoire discovery are quite interesting and can help understand the ability to capture details at finer granularities than the already fine-grained species identification.

**Weaknesses:**

Overall, I believe this work has the potential to be a strong contribution to the community and commend the authors’ effort. However, I think there is still room for improvement in several key areas before publication.


**\[Novelty\]**

1. The technical novelty of the paper is relatively low. Separately, self-supervised learning has been explored by BirdMAE, mixing bioacoustic data with general audio data has been explored by Perch 2.0 \[1\] (although this is more of a concurrent work), and effects of including non-avian species have also been explored by iNatSounds and SurfPerch.

**\[Standardized benchmarking\]**

2. For BirdSet, if I understand correctly, the authors use train splits for each datasets and learn linear probes. Given that the training dataset includes XenoCanto, I suspect that most if not all species in BirdSet tasks should be a part of the training dataset. Can the authors also report performance with the classifier head learnt during post-training? I believe the latter was used by the BirdSet authors, and a consistency in evaluation can be helpful for future work. These extra numbers could just go in the supplementary if the authors would like to preserve the current flow.

3. For BEANS classification tasks, if I understand correctly, the authors use “time-averaged embeddings”, but BEANS benchmark uses the first window only. I think the authors’ choice is more reasonable, but again argue in favour of consistency in evaluation. I understand that the baseline results presented by the authors are consistent with each other, but it would be helpful for future work to have easily comparable results.
4. For BEANS classification tasks, looking at the supplementary table 8, it seems like the authors include ESC-50, but not speech commands. This choice affects the mean-BEANS metrics and should be justified in the main text. ESC-50 is also missing in Table 5-6.

**\[Results\]**

5. All rows in Table 3 are repeated as-is in Table 4\. This seems unnecessary to me. I ask that the authors either keep them in a single table or remove the rows from Table 4\. I would recommend the former. I understand the benefit of the distinction of these results, but perhaps the authors can use a different background color or some other way to highlight the Table 3 numbers.
6. The authors choose to use WABAD as part of the training in an ablation. Since this is a densely labelled soundscape dataset, I believe this could be more useful as an evaluation dataset instead.
7. In Table 8, I was looking at “sl-BEATs-bio” and “sl-BEATs-all”. It seems like using AudioSet helps with ESC-50 considerably. Given the smaller aggregate improvement in Table 4, I take this to mean that including Audioset actually worsened the performance for the aggregate of the rest of BEANS classification tasks, which are the actual bioacoustics tasks. This seems counter to the claims of the benefit of training on general audio. The considerable improvement in ESC-50, which contains environmental sounds, may be partly due to similar sounds in AudioSet and may actually be misleading as an evaluation for bioacoustics tasks.
8. I am unclear about the motivation for Table 2b. As per my understanding, while the BEANS classification tasks have focal recordings, they still have a domain shift with regards to the species being considered. For example, aquatic species in watkins and insects in HumbugDB. The detection tasks also have this “label shift”, but for different species. To me, this is a confounding factor and I am unclear if the difference in behavior for the two can be attributed to the focal vs soundscape domain shift. Following the previous point, if ESC-50 is included in classification tasks, this is another confounding factor when assessing the effect of adding AudioSet.

**\[Minor Points\]**

9. Elements of the paper are going out of the margin in a few places (Table 2, Fig 4 and 5\)
10. Fig 2: font size of text is too small
11. Line 316 “ADD REF TO AGILE PAPER” :)
12. Line 213: I believe Perch is trained from scratch instead of starting from ImageNet. Not critical to the discussion, but may be worth noting.
13. The notation in Table 4 is confusing. What is the prefix “sl-E…” in sl-EAT-all$^{SL-SSL}$?
14. Notation (superscripts) is different in Table 8 vs Table 4

**References**

\[1\] van Merriënboer, Bart, et al. "Perch 2.0: The Bittern Lesson for Bioacoustics." arXiv preprint arXiv:2508.04665 (2025).

**Questions:**

1. I am curious about the Individual ID and Vocal repertoire discovery tasks. In the post training stage, different call types and individuals of the same species should have been assigned the same label, so I imagine the model should be learning similar embeddings for different individuals and calls of the same species. How do the authors explain this “emergent” ability? Is self-supervised learning the main contributor? Although for Individual ID, Perch and BirdNET seem to be doing on par and maybe even better without any self-supervision.
2. The takeaways about the benefits of self-supervised learning are counter to the observations in Perch 2.0 \[1\]. Could the authors suggest possible factors to explain this difference.

---

> ### Author Response · Authors · 2025-11-21
>
> We thank the reviewer for their comments, particularly for raising useful questions which we addressed in the paper. The reviewer valued the extensiveness of our evaluation framework, unifying different evaluation strategies proposed in the literature, the training recipes, and the newly curated tasks and datasets. We take the following paragraphs to answer the questions and address the improvements along the raised issues, particularly on the novelty, benchmarking, and results.
>
> **Novelty**
> \[1\] In Section 1 lines 59-70 and in Section 2 we contrast how current work in SSL, data-mix and cross-taxa transfer contrasts with this paper. Namely, these topics have been previously explored but in a limited context (training paradigm, data, benchmarks) and also not simultaneously. For instance, SSL has been investigated by BirdMAE but trained and evaluated solely on birds and not combined with SL.
>
> **Benchmarking**
> \[2\] We included the BirdSet post-training evaluation in Section C5 in the Appendix and we also addressed this as a general comment to all the reviewers.
> \[3\] BEANS uses a short window from the beginning of each audio clip, with a different length per dataset (e.g. 10 seconds for DCASE , 2 seconds for Watkins, etc.). We match this exact setup and these windows exactly, so we are consistent with BEANS.
> \[4\] We have added Section C4 in the Appendix to address the impact of excluding ESC-50 from averaging the results. We addressed this as a general comment to all the reviewers.
>
> **Results**
> \[5\] Following the reviewer’s recommendation we kept the aggregated results table and shaded the rows corresponding to the EfficientNet models.
> \[6\] We agree that the size of WABAD is comparable to some of the detection datasets in the benchmark. However, we wanted to see what is the impact of using soundscape data in training because most of the training data is focal.
> \[7\] We included Section C4 in the Appendix to remove the ESC-50 in the aggregate results. Please see the section about it in the general comments above, including the discussion about the gains of using general audio (the gains still stand on non-focal data, particularly the detection tasks).
> \[8\] We have added a paragraph to clarify the evaluation setup in Section 4.2. Going beyond domain shifts in taxa, recording conditions (focal vs soundscape), there are multiple confounders that can affect these results. With this experiment we wanted to see how good the raw representations of the models are by using a training free metric (R-AUC). Performance metrics obtained from probing may show different trends because they are more sensitive to learning those confounding factors. However, we agree that a controlled experiment where noise, species distribution, and other factors are manipulated within the same dataset would give a better answer. We are not aware of any existing dataset to allow for this experiment.
>
> **Minor issues**
> \[9-12\] We did these corrections in the paper.
> \[13\] Table 2 provides a legend comprising each checkpoint name. According to this legend the sl-EAT is the checkpoint resulting from post-training an EAT model.
> \[14\] We add the SL, SSL, SL-SSL superscripts to the tables containing full results.
>
> **Questions**
> \[Question 1\] We note that for the Individual ID and Repertoire, SL and SL-SLL models are superior. We hypothesize that, similarly to Perch 2.0 (Table 7 in the Appendix), SL on a high number of classes learns the priors to discriminate with high granularity between different categories of sounds and this transfers to Individual ID and Repertoire.
> \[Question 2\] We have offered an answer to this question in the general comments, point 5\. Namely, the findings we discuss in Section 4.2 are not contradictory but complementary: SSL pretraining helps with out-of-distribution tasks, so SSL+SL may represent a better solution than SL alone particularly when dealing with detection tasks.

---

> > ### Author Response · Authors · 2025-11-27
> >
> > We hope that our prior response and the updated revision have thoroughly addressed the concerns raised. We remain open to any further discussion should additional points require clarification.

---

> ### Comment · Reviewer_B3CZ · 2025-11-27
>
> Thank you for your detailed response. Several of my concerns have been addressed. I appreciate the added BirdSet post-training evaluation and clarifications about the benchmark configurations, results tables and evaluation setup.
>
> I had a few follow up questions and comments.
>
> **[Re: 1]** I agree with the authors about their contribution as a simultaneous exploration in a broad bioacoustic context. However, I view this as a benchmarking and standardization contribution rather than added technical novelty. I still believe this is valuable work, but retain my concern about technical novelty.
>
> **[Re: 2]** Computing BirdSet-average results in Table 11, I see that the bio-only model has an average of 33.4% while the all dataset model has an average of 32.8%. This is the opposite trend of the probing results in the main table. Could the authors comment on this? Does adding general audio lead to better features but a worse classification head?
>
> **[Re: 4]** Thank you. Please also add ESC-50 in Table 4.
>
> **[Re: 7]** Thank you. Which setting do you plan to keep in the main table, w/ or w/o ESC50? Could you also please provide a version of Fig 2b without ESC-50?
>
> **[Re: 8]** In my earlier comment, I meant Fig 2b and not Table 2b, apologies for the confusion. I agree with the authors about the difficulty in removing confounding factors and that current datasets don't allow a controlled experiment. Which is why I think that a direct comparison of BEANS classification and detection results in the scatter plot here is not meaningful, even with the training free metric. I may yet be misunderstanding the objective of this figure, and hope the authors can clarify how this figure helps support or augment the claims of the paper.
>
> **[Re: Question 2]** My general takeaway from Perch 2.0 was that bioacoustics training data has reached a stage where simple supervised learning is hard to beat, and that SL-SSL leads to marginal benefits over SL. This paper seems to suggest that SL-SSL is still meaningfully better than SL, which is the counter observation I was referring to. Following points raised by reviewer WxmL, it looks like this paper also does not explicitly compare SL vs SL-SSL for the same architectures, something that I had missed before.
>
> ---
> **New concerns**
>
> **[15]** Regarding the bio vs all training, switching from DT to LT+ reverses the trend for BirdSet and excluding ESC-50 leads to a significant drop in improvement (1% to 0.3%). This raises new concerns to me about the actual benefit of adding general audio. More broadly, it also raises the question about the sensitivity to the choice of the evaluation setup. How much does the evaluation setup matter in bioacoustic encoding? Are the metric and evaluation dataset choices made by the authors the "right" ones?

---

> ### Author Response · Authors · 2025-12-02
>
> \[Re: 1\] Besides benchmarking and standardization we propose a few models e.g. sl-BEATs resulting from our training recipes that are SotA on multiple tasks.
>
> \[Re: 2\] We assume that the sl-BEATs-bio model trained on mostly bird data (XC, iNat etc.) with strong mixup and noise augmentations is already a strong baseline for bird detection in soundscape recordings and adding additional sound event classes may not help the classifier but it acts as a strong regularization increasing the robustness of the representations.
>
> \[Re: 4\] Added ESC-50 to Table 4\.
>
> \[Re: 7\] We have already added a paragraph in Section 4.3 making clear that ESC-50 is part of benchmarking, considering it is still useful for habitat classification and other tasks relevant for conservation. We will keep the non-ESC-50 results in the Appendix. A version of Fig2B without ESC-50 has been already added, Figure 7 in the Appendix, and it does not change the message.
>
> \[Re: 8\] As stated in the revised pdf, section 4.2, “*For this comparison we want to see how discriminative raw representations from the models are with respect to the target datasets, and we use ROC AUC as a training-free metric. Compared across benchmarks, this analysis should give an idea about how generalizable the embedding space is. In this analysis we trade-off experimental control for scale, using large datasets. Running the same analysis on datasets*
> *where we control for species distribution, noise conditions, and other confounders should give a*
> *better explanation on how robust these representations are. Because we are not aware of any dataset that offers this controlled conditions we leave this for future work.*”
>
> \[Re: Q2\] To our best knowledge, Perch 2.0 does not include a comparison with an SL-SSL model. Thus the findings are not contradictory but complementary. One common finding is that SSL alone is inferior to both papers. With respect to including a comparison with a pure-SL method within a transformer, we are open to include that in the final paper. However, we argue that starting from a random checkpoint vs a pre-trained one, as we do now, would not significantly change the message of the paper i.e. we would not note better results for the pure SL method. We base this assumption of the current findings in the literature \[1,2\] which found that using a random initialization leads to inferior results. In fact, starting from scratch is something that researchers would do pre-SSL-era when they were developing models for single datasets and not when evaluating representations e.g. emotion recognition \[3\].
>
> \[Re: 15\] First, BEANS is solely one of the benchmarks we evaluate on, adding the general audio has benefits on other benchmarks (BirdSet, ID, Repertoire). Second, in the case of BEANS, we note that the ranking of the models has not changed as a result of excluding ESC-50 from the evaluation, although there are valid conservation-related reasons for including it. The sl-BEATs model we propose are still the best models on multiple tasks. As a conclusion, we believe that the message remains unchanged.
>
> A core contribution of this paper and a limitation of the literature in bioacoustics encoding is standardizing the evaluation setup: we report a wide set of metrics related to probing, retrieval, and clustering on two existing benchmarks and adding two additional benchmarks (id and repertoire), encompassing all previous evaluations protocols as special cases.
>
> \[1\] Koutini, K., Schlüter, J., Eghbal-Zadeh, H., & Widmer, G. (2021). Efficient training of audio transformers with patchout. arXiv preprint arXiv:2110.05069.
> \[2\] Alonso-Jiménez, P., Serra, X., & Bogdanov, D. (2023). Efficient supervised training of audio transformers for music representation learning. arXiv preprint arXiv:2309.16418.
> \[3\] Akinpelu, S., Viriri, S., & Adegun, A. (2024). An enhanced speech emotion recognition using vision transformer. Scientific Reports, 14(1), 13126\.

---

### Official Review · Reviewer_Jxj8 · 2025-11-02

**Soundness:** 2
**Presentation:** 3
**Contribution:** 2
**Rating:** 4
**Confidence:** 4

**Summary:**

The paper is an empirical study of bioacoustic encoder. The paper explores several benchmarks and training configurations.

**Strengths:**

The overall goal of studying bioacoustics is fascinating and important. I think it is a great application domain for the NeurIPS community.

**Weaknesses:**

The contribution is interested by limited in several ways. The reported results will be interesting for the community working on this area, but might not be interesting for a wider set of researchers. An experimental paper would need to introduce challenges derived from bioacustics that can motivate technical advances, or introduce a new dataset, or a better definition of the task with new evaluation metrics, or solving a new problem. But the current contribution is limited along all those dimensions.

The technical novelty is limited.

Some of the results are not very surprising such as
- adding general audio into the training significantly improves model transferability
- self-supervised models under-perform supervised models
Both those results are not unexpected.

The approach uses standard evaluation metrics (linear probing, retrieval, and clustering) and tasks (classification and detection) used by the community. Although some of those are extensions of existing benchmarks, on its own are a limited contribution.

**Questions:**

Would it be possible to better articulate if there are contributions along these questions and what those contributions are?
- what new technical advances are needed?
- Is there a new dataset being introduced?
- Is there a better definition of the task being proposed?
- Are there new evaluation metrics being introduced?
- Is there a new problem/task being introduced?

---

> ### Author Response · Authors · 2025-11-21
>
> We thank the reviewer for their questions. We answer each one of them in the following paragraphs.
>
> **Limited contributions**
> We introduce the main contributions in Section 1 lines 72-90. We summarize the limitations of the current research in bioacoustics in Section 1 lines 63-72 and in Section 2, namely the lack of standardization of training recipes and models, expanding the training and benchmarking data beyond birds, and with this paper we aim at addressing these limitations. In addition Table 4 in the Appendix summarizes the differences with respect to the existing papers regarding training data and benchmarks.
> Regarding the findings being interesting for a larger ML community, bioacoustics is an interesting ground for many machine learning problems and recently a few bioacoustics papers were presented at large ML conferences \[1,2,3\].
>
> **Questions:**
>
> - “New technical advances” \- Yes, there are new technical advances. The proposed training recipes based on combining SSL and SL achieved state of the art on the benchmarks across different architectures (CNNs and transformers).
> - “New dataset” \- Yes, there are new datasets. We add more pre/post-training data than existing bioacoustic research. We propose new evaluation benchmarks for individual id and repertoire discovery, tasks that were previously not studied in representation learning for bioacoustics (see Table 4 in the Appendix)
> - “Better task definition”: Yes, we propose a better task definition. We go beyond species classification and detection and we test how well representations transfer to individual id and vocal repertoire.
> - “New metrics” \- Yes, there are multiple metrics considered together, something not common in bioacoustics. We comprehensively evaluate all datasets across the same metrics related to probing, clustering, and retrieval, whereas previous research solely considered one of the metrics classes.
> - “New problem/task” \-  In fact, this is one of our main contributions, we formalize a training and evaluation paradigm comprising SSL, SL, SL+SSL, across different training data (general audio, bioacoustics) and 26 benchmarks.
>
> \[1\] Boudiaf, Malik, et al. "In search for a generalizable method for source free domain adaptation." ICML (2023).
> \[2\] Rauch, Lukas, et al. "Birdset: A large-scale dataset for audio classification in avian bioacoustics." ICLR (2025).
> \[3\] Robinson, David, et al. "NatureLM-audio: An audio-language foundation model for bioacoustics." ICLR (2025).

---

> > ### Author Response · Authors · 2025-11-27
> >
> > We hope that our prior response and the updated revision have thoroughly addressed the concerns raised. We remain open to any further discussion should additional points require clarification.

---

### Author Response · Authors · 2025-11-21
**General comments to the reviewers [Part1]**

We would like to thank the reviewers for their valuable comments. Before addressing the individual comments, we address specific issues that either required more experiments or were common to multiple reviewers. We have enhanced the paper with these new experiments and added 3 new sections to the Appendix, C4-6, to include these new results. Following reviewer’s B3CZ recommendation we merged Tables 3 and 4 and all the references to tables in the rebuttal refer to the new table indices. We have increased readability of the figures, added paragraph headings to longer sections, and fixed the minor issues in text. We have marked the changes with blue font color.

**New experiments**

1. **Missing SSL models, the choice of SSL models (reviewers WxmL,ykFi)**.
   We included BirdMAE (with the BirdMAE-Huge checkpoint) as another bioacoustic SSL suggested by reviewer WxmL. All the results tables and plots were modified accordingly and now we have BirdMAE as a strong baseline for multiple tasks particularly the mosquitos (HBDB) and bats datasets in BEANS.

   Regarding the choice of SSL models (reviewer ykFi), as we describe in Section 3.4 we are not focusing solely on BEATs but evaluating a wide range of pretrained SSL models from general audio (BEATs, EAT) and bioacoustics (AVES, BirdMAE). Moreover, we pretrain our own EAT model in addition to the pretrained EAT. Note the SSLAM model official implementation [https://github.com/ta012/SSLAM](https://github.com/ta012/SSLAM) is mostly an additional iteration on the EAT codebase.

2. **Effect of ESC-50 in the evaluation, BEANS auxiliary datasets (reviewer B3CZ)**. The BEANS benchmark contains two auxiliary non-bioacoustics datasets ESC-50 and SpeechCommands. We excluded SpeechCommands because this dataset is irrelevant for the bioacoustics applications and speech is a well researched area beyond bioacoustics. In contrast we included ESC-50 because general sound classification is useful for some conservation tasks like habitat classification, poaching monitoring (gunshots, explosions) and this is the reason why we included it in the aggregated tables.  We added a clarification in Section 3.4.

   Following the reviewer’s B3CZ comments, and to disentangle the effects of the data mix and answering the question: "does including general sound in training give better representation?" we report the averaged results without ESC-50. We include the original tables side-by-side for comparison in a new section in the Appendix (the tables in the Section C4).

   We note that the ranking per-model has not changed e.g. sl-BEATs models are better than EffNetB0 models. However, there is less of a gap between *bio* and *all* setups, with *bio* being slightly better in some cases such as EffNetB0.

3. **BirdSet direct post-training evaluation (reviewer B3CZ).** We take the SL checkpoints we trained for the post-training phase and we directly evaluate them on the BirdSet dataset. Because BirdSet training classes are identical to the test ones, and our models consider a large number of disjoint classes outside the BirdSet taxa, we restrict the output of our models by  considering solely the logits corresponding to the datasets in BirdSet. This evaluation setup is comparable to the LT (large training) setup in the BirdSet paper with the following additions: (1) more training data and (2) fine-tuning a whole model, hence we call it LT+. It contrasts with the DT (dedicated training) setup which we reported initially i.e. linear probing of a model on each BirdSet subset. Notably, in our case DT is done on top of LT+ and it seems to forget some of its performance. Similarly to the BirdSet paper the LT, and aforementioned LT+, have better results than DT. The results are presented in the Appendix Section C5, Table 13\.

   Why is LT+ better than DT for BirdSet? Both of the setups contain a domain shift and although we add the same augmentations in LT+ and DT, DT seems to overfit to often small training set originating in Xeno-Cantto, whereas LT+ sees more data, and learns to discriminate with high granularity a high number of classes by learning the time-frequency priors, useful for the domain shift.

4. **Alternatives to linear probing (reviewers WxmL,ykFi).** We ran a small-scale ablation (50 epochs) to compare a linear probe head with an attention probe head for a variety of models. The results are included in the Appendix Section C6, Figure 7\. Generally, we note that pretrained models (transformers) benefit from the attention head and the ranking of the models slightly changes with less of a difference between the best performing and the worse performing model.

---

> ### Author Response · Authors · 2025-11-21
> **General comments to the reviewers [Part2]**
>
> **Clarification**
>
> 5. **The results are not very surprising (reviewers Jxj8,WxmL) vs different findings to Perch 2.0 (reviewer B3CZ)**
>
> While the reviewers Jxj8,WxmL found that the findings are not surprising, the reviewer B3CZ found that the findings are somehow surprising when compared to a concurrent work (Perch 2.0). We address all these claims.
>
> *Re: it is not surprising that general audio improves model transferability.* The question of what representations learned on what data transfer to what tasks and under which training paradigms is not trivial nor has been previously studied at this scale in bioacoustics: same training conditions for multiple encoders and cross-taxa evaluation. We give a detailed comparison with the previous methods in Section 2\. The results and our discussion are more nuanced than “general audio \+ bioacoustics is better”. For instance, when looking at the results per dataset in Tables 6-9 we note that while the model trained on general audio \+ bioacoustics (ending in “-all”) work better for the detection tasks in BEANS and BirdSet and surprisingly for Individual ID and Repertoire (except sl-EAT), the models trained solely on bioacoustics (ending in “-bio”) work better for the classification tasks in BEANS. We do further data ablations in Figures 4 and 5 to better understand this cross-taxa i.e. answering the question “does excluding some species improves certain tasks?”.
>
> *Re: it is not surprising that self-supervised models under-perform supervised models.* Our findings and discussion are more nuanced than this (Section 4.2). What we found is that the combination SL-SSL is the one that works best and pretraining helps. To add to that, we have included BirdMAE, a strong SSL benchmark particularly for unseen taxa such as insects and bats. To that extent, there are pure SSL models or pure SL models that work well for particular datasets but are not best overall. Moreover, the new probing experiments we included in the appendix show that SSL models become competitive or better than the SL models for detection tasks when replacing the linear probe head with an attention probe head.
>
> Re: *The takeaways about the benefits of self-supervised learning are counter to the observations in Perch 2.0.* The findings we discuss in Section 4.2 are not contradictory but complementary: SSL pretraining helps with out-of-distribution tasks, so SSL+SL may represent a better solution than SL alone particularly when dealing with detection tasks.
>
> 6. **Running seeds, statistical significance (reviewers WxmL,ykFi)**
>
> We evaluated 20 models x 26 datasets for a single fixed random seed that controls solely the random initialization of a simple linear layer on top of the averaged embeddings. We admit that there may be some variation to the results but it should be smaller than fully-finetuning the whole model. This variation solely happens for probing and not for retrieval and clustering where we use training-free metrics. We use fixed/official dataset splits for the benchmark datasets i.e. not controlled by the random seeds. These experiments took a few weeks to complete and running multiple seeds would have had a prohibitive computing cost for us.
>
> 7. **Open sourcing, reproducibility**
>
> We are currently working towards building an API to facilitate access to the checkpoints, models, probe heads. training and fine-tuning of bioacoustics encoding. We will open source the codebase for this paper along with that. This should happen within the next 4 months.

---

### Meta-Review · Area_Chair_9mox · 2026-01-08

**Summary:**

This paper introduces a large-scale empirical study on deep learning for bioacoustics.  It studies various architectures, training configurations and benchmarks on the area of bioacoustics.

Overall, the reviewers appreciate the author efforts in their work, as the area of bio-acoustics is both interesting and important for ML.

The reviewers share many common concerns:
(a) limited technical novelty
(b) experimental rigorousness
(c) clarity and presentation, with questions

**Reviewer Concerns:**

The rebuttal addresses most issues on clarity and presentation, and makes attempts to justify their experimentation and benchmarking choices.  However, it is not able to address the point on technical novelty.

It is not unexpected (and this is acknowledged by the reviewers) that the technical novelty of such a paper is low.  While this does limit the overall contribution, the AC believes it is sufficient to pass the bar for publication, given the extensiveness of the benchmarking and experimentation.

**Reviewer Scores:**

Jxj8 - this reviewer initially gave a 4.  I don't think this reviewer would have changed their score, given that their overall assessment of the paper was that they felt it lacked sufficient technical contribution.  The author responses treat the questions superficially without addressing the overall concern.

B3CZ - this reviewer gave a 6.  Most concerns were clarification based on the experimental protocols and addressed by the rebuttal.  I think this reviewer would keep or raise their score.

WxmL - this reviewer gave a score of 6.  I think they would've retained their score but not change.

ykFi - this reviewer gave a score of 4.  They were semi-positive from the responses and mentioned weighing a decision to raise the score to 6  in the discussion.

---

### Decision · Program_Chairs · 2026-01-26

Accept (Poster)